



# Variability of surface climate in simulations of past and future

Kira Rehfeld[1], Raphaël Hébert[2], Juan M. Lora[3], Marcus Lofverstrom[4], and Chris M. Brierley[5]

[1]Institute of Environmental Physics, Ruprecht-Karls-Universität Heidelberg, INF 229, 69120 Heidelberg, Germany
[2]Alfred-Wegener Institute Helmholtz-Center for Polar- and Marine Research, Telegrafenberg A45, 14473 Potsdam, Germany
[3]Department of Geology and Geophysics, Yale University, 210 Whitney Ave, New Haven, CT 06520, US
[4]Department of Geosciences, University of Arizona, 1040 E. 4th Street, Tucson, AZ 85721, US
[5]Department of Geography, University College London, London, WC1E 6BT, UK

**Correspondence:** Kira Rehfeld (krehfeld@iup.uni-heidelberg.de)

**Abstract.** It is virtually certain that the mean surface temperature of the Earth will continue to increase under realistic emission scenarios. Yet comparatively little is known about future changes in climate variability. We explore changes in climate variability over the large range of climates simulated by the Coupled Model Intercomparison Project Phases 5 and 6 (CMIP5/6) and the Paleoclimate Modeling Intercomparison Project Phase 3 (PMIP3). This consists of time slices of the Last Glacial Maximum, the Mid Holocene and idealized warming experiments (1% $CO_2$ and abrupt4×$CO_2$), and encompasses climates with a range of 12°C of global mean temperature change. We examine climate variability from different perspectives: the local interannual change, coherent climate modes and through compositing extremes. The change in the interannual variability of precipitation is strongly dependent upon the local change in the total amount of precipitation. Meanwhile only over tropical land is the change in the interannual temperature variability positively correlated to temperature change, and then weakly. In general, temperature variability is inversely related to mean temperature change - with analysis of power spectra demonstrating that this holds from intra-seasonal to multi-decadal timescales. We systematically investigate changes in the standard deviation of modes of climate variability, such as the North Atlantic Oscillation, with global mean temperature change. While several modes do show consistent relationships (most notably the Atlantic Zonal Mode), no generalisable pattern emerges. By compositing extreme precipitation events across the ensemble, we demonstrate that the atmospheric drivers dominating rainfall variability in Mediterranean climates persist throughout palaeoclimate and future simulations. The robust nature of the response of climate variability, between both cold and warm climates and across multiple timescales, suggests that observations and proxy reconstructions could provide a meaningful constraint on climate variability in future projections.

## 1 Introduction

Slow and sustainable changes in mean climate conditions are important to understand climatic risks and uncertainties (IPCC-AR5, 2013). However, understanding changes in the variability around this mean is at least as pressing as the understanding of changes in mean climate for society and agriculture (Katz and Brown, 1992). This is because societal (Alexander and Perkins, 2013; Katz and Brown, 1992; Hsiang et al., 2013) and ecosystem (Seddon et al., 2016; Stenseth, 2002) impacts scale with





climate variability, and increasing variability leads to increasing extreme events (IPCC-AR5, 2013; Schär et al., 2004).

Climate variability can be defined as variations in the mean state and other statistics (e.g. standard deviations, the frequency of occurrence of extremes) of temperature, precipitation and atmospheric circulation on spatial and temporal scales beyond individual weather events (Qin et al., 2014; Xie et al., 2015). Internal variability arises due to complex (often nonlinear) internal processes within the atmosphere-ocean-biosphere-cryosphere system (Deser et al., 2012a; Olonscheck and Notz, 2017), or as forced variability in response to changes in natural or anthropogenic forcing (Foster and Rahmstorf, 2011). However, the actual evolution of climate combines anthropogenic forcing and natural climate variability (Horton et al., 2016), with internal variability dominating the local-to-regional synoptic evolution (e.g., Deser et al., 2012a; Wallace et al., 2015). A core focus of research has been the investigation of major climate phenomena, *modes of variability* (Qin et al., 2014) such as the El-Niño/Southern Oscillation (Walker and Bliss E.W., 1932; Bjerknes, 1966), and their contemporary change and representation by climate models (Deser et al., 2010, 2012a; Phillips et al., 2014). Their projected changes, and relevance for future regional climate evolution remain uncertain (Xie et al., 2015; Christensen et al., 2013). At the same time, atmospheric circulation changes contribute strongly to internal climate variability and, inherently, uncertainty of future projections (Thompson et al., 2015).

Trends established based on the instrumental record are uncertain, and both increasing (Hansen et al., 2012) or decreasing (Rhines and Huybers, 2013; Lenton et al., 2017) trends in temperature variability have been established. These trends differ amongst world regions (Rhines and Huybers, 2013; Huntingford et al., 2013): More economically underdeveloped areas were found to be more affected by increases in temperature variability than the more high-latitude developed regions (Bathiany et al., 2018). In any region, damages do, however, scale with increased variability (Katz and Brown, 1992; Alexander and Perkins, 2013). Therefore, there is a need to better understand changes to climate variability under warming. A warming similar to that projected for the next centuries (IPCC-AR5, 2013) occurred from the Last Glacial Maximum (LGM, 27-19 thousand years before present, 27-19 kyrs BP) before apparently stable Holocene climate conditions were reached (since 11.7 kyrs BP). Along with this warming, a reduction in centennial to millennial-scale temperature variability to 1/4th the Glacial level was estimated based on palaeoclimate proxy data and linked to the reduction of the local meridional temperature gradients (Rehfeld et al., 2018). Based on this mechanistic link, a continued decrease in temperature variability at the global scale could be expected at long timescales (Rehfeld et al., 2018). It is, however, unclear how these long timescales link to the synoptic to decadal variability, which is not generally observable with palaeoclimate proxies. There is corroborating evidence based on model simulations for decreases in variability at interannual (Holmes et al., 2016) and longer (Brown et al., 2017) timescales. In particular, the contemporary decline in Arctic sea-ice extent has been linked to declines in temperature variability at a global scale (Huntingford et al., 2013; Olonscheck and Notz, 2017; Bathiany et al., 2018). At the seasonal scale, higher temperature variability over Northern Hemisphere (NH) land in summer (Holmes et al., 2016) has been observed, consistent with increases in summer extremes (Coumou and Rahmstorf, 2012; Pfleiderer et al., 2019).





Clearly, changes in hot temperature extremes are linked to the local mean temperature change (Rhines and Huybers, 2013), but increasing synoptic variability could contribute to more frequent heat waves (Horton et al., 2016) and circulation changes to larger winter temperature variability (Screen and Simmonds, 2014) and persistence of weather patterns (Francis and Vavrus, 2012). Increasing precipitation, and precipitation variability, have been linked to warming (Pendergrass et al., 2017; Collins

et al., 2013; Allen and Ingram, 2002; Held and Soden, 2006). In most climate models, precipitation variability was found to increase over land for future warming scenarios, with variability increasing at the same or a higher rate than the mean (Pendergrass et al., 2017). Precipitation changes are, however, strongly linked to changes in circulation and internal variability, which are not fully understood (Hawkins, 2011; Christensen et al., 2013; Deser et al., 2012a).

Here we investigate the linkage between mean-state and variability changes of temperature and precipitation across a wide range of global mean temperatures. In particular, we examine changes in climate variability and on interannual to multidecadal timescales in simulations conducted in the framework of the Palaeoclimate Modeling Intercomparison Project phase 3 (Braconnot et al., 2012, PMIP3), as well as the Coupled Model Intercomparison Project phase 5 (CMIP5, Taylor et al., 2012) and phase 6 (CMIP6, Eyring et al., 2016a) it is affiliated to. We contrast changes in interannual variability across simulations for

the LGM (denoted *lgm* in the following), the mid-Holocene (*midHolocene*, 6kyrs BP) and for the idealized warming scenarios with 1%CO$_2$ increase per year (*1pctCO2*) as well as an abrupt quadrupling of CO$_2$ (*abrupt4xCO2*). Section 2 gives details on these experiments, the data preprocessing and the comparison metrics. Section 3 examines changes in local interannual variability, modes of variability, the drivers of extreme precipitation changes and in the spectrum of variability. In Sect. 4 we discuss how this compares to previous findings, and identify key uncertainties. We conclude, in Sect. 5 with a discussion on

prospects for validation of modeled climate variability.

## 2  Data and Methods

### 2.1  Model simulations

The core aim of this study is to compare past and future climate simulations, and to assess the similarity – or differences – in climate variability across different Earth system states. We consider a range of state-of-the-art climate models (listed in Table

1). Therefore, this analysis is based on climate model experiments coordinated by the Coupled Model Intercomparison Project (CMIP) phase 5 (CMIP5; Taylor et al., 2012) and phase 6 (CMIP6; Eyring et al., 2016a) as well as the corresponding Palaeoclimate Model Intercomparison Project phase 3 (PMIP3; Braconnot et al., 2012). There are 25 climate models considered in this study (Table 1 and Fig. 1). The preindustrial control (*piControl*) simulations represent constant preindustrial (PI) conditions and are the baseline for comparison in all our analyses. We analyze the air surface temperature ('tas'), precipitation ('pr'), sea

surface temperature (SST) and sea-level pressure (SLP) variables.





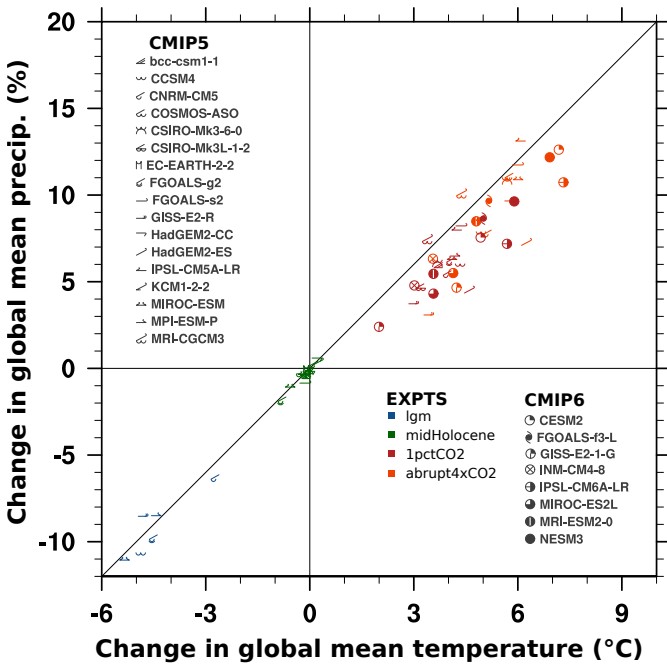

**Figure 1.** Hydrological sensitivity across the past and future model ensemble. The change in global mean temperature from the PI is plotted against the percentage change in global mean precipitation rate. Symbols indicate the different climate models, following Table 1. Colours show the different experiments. The line indicates 2% change in precipitation per Kelvin temperature change.

## 2.2 The Last Glacial Maximum experiment *(lgm)*

The last glacial maximum (*lgm*) experiment represents conditions of 21,000 years ago. Globally averaged surface temperature was about 3-5 degrees colder than today (Annan and Hargreaves, 2013; Shakun and Carlson, 2010) in response to a global mean radiative forcing of $\sim$ -4W/m² (Broccoli, 2000) by reduced greenhouse gas concentrations (GHG), large continental
5  ice-sheets, and a low sea-level (Clark and Mix, 2002; Broccoli, 2000; Annan and Hargreaves, 2015). A standard set of forcings (orbit, GHG) and boundary conditions (ice sheets) was set out in PMIP3 (Braconnot et al., 2012; PMIP3, 2010) and PMIP4 (Kageyama et al., 2018). In particular, the ice sheet extent and height is modified with respect to the *piControl* configurations, to reflect the extensive LGM Northern hemisphere ice sheet cover. $CO_2$ concentrations are fixed at 185ppm, $CH_4$ at 350ppb and $N_2O$ at 200ppb (PMIP3, 2010), whereas solar constant, vegetation and aerosols follow the preindustrial control setup (Taylor
10  et al., 2012). Overall, insolation was higher-than-preindustrial in winter in both hemispheres, and lower-than-preindustrial summer in both hemispheres (up to -12 W/m² in NH high latitudes) (Otto-Bliesner et al., 2006). This corresponds to a reduced seasonal contrast in the top-of-atmosphere radiation. The multi-model mean, shown in Fig. 2a, shows global cooling, but strongest cooling in the polar regions and above ice sheets.





## 2.3 The mid Holocene experiment (*midHolocene*)

The *midHolocene* experiments represent conditions of 6,000 years, during the peak warmth of the current interglacial (Taylor et al., 2012; Braconnot et al., 2012). The different orbital configuration with higher-than-present-day obliquity and eccentricity lead to an enhanced seasonal contrast in insolation, with stronger insolation in June to September from the high northern latitudes down to 30S (up to $32W/m^2$ in NH summer), stronger insolation in September to November/ SH spring ($+48W/m^2$) (30S to 90S), and negative insolation anomalies of similar magnitude in the other montHS of the year (Otto-Bliesner et al., 2006). This lead to a weak global mean insolation anomaly. Greenhouse gas concentrations in the PMIP3 ensemble were prescribed as for the *piControl* simulation ($\sim$ 280 ppm $CO_2$, 650ppb $CH_4$, 270 ppb $N_2O$), as were the configurations of vegetation, aerosols, ice sheets, topography and coastlines(PMIP3, 2010). In previous model intercomparison exercises, global mean temperatures were found to be similar to today (Otto-Bliesner et al., 2006), but proxy data from the Northern Hemisphere support warmer temperatures (Wanner et al., 2015; Marcott et al., 2013).

## 2.4 The warming experiments *1pctCO2* and *abrupt4xCO2*

To complement the palaeoclimate simulations, we analyze two baseline experiments each model in CMIP5 and CMIP6 has performed: the idealized warming experiments *1pctCO2* and *abrupt4xCO2* (Taylor et al., 2012; Eyring et al., 2016a). In the *abrupt4xCO2* experiment $CO_2$ concentrations are abruptly quadrupled from preindustrial conditions to analyze fast feedbacks and climate sensitivity (Eyring et al., 2016a). The simulations are continued for at least 150 years. We analyze the years 100-150 for all simulations. [Note that we follow the naming scheme of CMIP5 (*abrupt4xCO2;* Taylor et al., 2012), while in CMIP6 the experiment name is *abrupt-4xCO2* (Eyring et al., 2016a). The experimental protocols are equivalent between the CMIP generations (Taylor et al., 2012).] The $CO_2$ concentrations in the *1pctCO2* simulations are prescribed to increase by 1% per year in a compound fashion starting from preindustrial conditions (Eyring et al., 2016a). The change in global mean temperature at the time of $CO_2$ doubling in this experiment is called the transient climate response (TCR; Andrews et al., 2012). This compound increase achieves a quadrupling of carbon dioxide after 140 years, but the climate system is still highly transient. The *1pctCO2* simulations are continued between 140-160 years, of which we analyze the final 50 years. The realized warming in the *1pctCO2* scenarios is less than in the *abrupt4xCO2* runs (Table 1), as the system is still farther from equilibration.

## 2.5 Preprocessing of model simulations

The model output is treated in a consistent fashion across all the analyses. We always analyze the 50 years of each simulation (the final 50 in all but the *abrupt4xCO2* experiment). For the variability analyses, output is converted to anomalies with respect to the monthly climatology over the 50 years using the `ncl` function `rmMonAnnCycTLL`. These anomalies are then linearly detrended at each gridpoint using the `ncl` function `dtrend_msg_n`. This process removes the changing mean state in the transient simulations and is based on the conventions of the Climate Variability Diagnostics Package (CVDP, Phillips et al.,





2014; Eyring et al., 2016b). For the analyses performed here, done at annual resolution, we do not apply the PaleoCalAdjust software to account for the calendar effects (Bartlein and Shafer, 2019).

## 2.6 Comparisons across the ensemble

All model output used in the study is available for download on the Earth System Grid Federation (Eyring et al., 2016b). Each
model is weighted equally during ensemble averaging. These experiments provide a large range of global mean temperature (GMT) changes (Fig. 1), ranges from -6 to +6.5 K with respect to the preindustrial state. Over this range of 12K in GMT, the area-weighted global mean precipitation (GMP) varies between -12% for the *lgm* experiments and +12% for the *abrupt4xCO2* experiments. The slope of the relationship between temperature change and precipitation change is known as the hydrological sensitivity (HS, O'Gorman et al., 2011). For CMIP5 and CMIP3 models, values between 2 and 3 % $K^{-1}$ have been established
(Li et al., 2013; O'Gorman et al., 2011; Allen and Ingram, 2002). Based on the mean temperature and precipitation values for each model (Fig. 1), we calculate HS individually for each GCMs (Tab. 1) and explore ensemble wide relationships (sections 3.1 & 3.2).

## 2.7 Diagnosing variability changes

This research spans across several different definitions of variability described in the literature. We term the kind of variability
analysed by e.g. Huntingford et al. (2013) and Pendergrass et al. (2017) as "local variability", in that it considers the year-to-year variations at an individual location. There has been a concerted effort to investigated the preferred spatial patterns and temporal variations that account for large-scale features in variance in the climate system. We term these as "modes of climate variability", and they are considered as the product of a specific spatial pattern and an associated index time series (Qin et al., 2014). They are diagnostic measures for teleconnections or surface climate patterns, and defined on pressure,
temperature or precipitation fields. Here, we investigate the changes of ten modes of variability from the detrended time series following the workflow of the Climate Variability Diagnostics Package (CVDP, Phillips et al., 2014). We investigate seven atmospheric-oceanic coupled modes defined as predominant SST patterns, and three extratropical atmospheric modes with large-scale teleconnection patterns (Deser et al., 2010; Nigam, 2003).

### 2.7.1 Local variability

Local variability is computed as the standard deviation of the annual mean temperature or precipitation. In all simulations a 50 year subset was selected (often the final 50 years, Sec. 2.5), anomalies with respect to the simulations climatology computed and then a linear trend removed. Variance ratios are computed on the individual model grids and interpolated bilinearly onto a common 1×1° latitude/longitude grid prior to ensemble averaging.



| Model | CMIP6? | ECS (K) | midHolocene | lgm | 1pctCO2 | abrupt4xCO2 | Hydro. Sens. ($\eta_a$, %/K) |
|---|---|---|---|---|---|---|---|
| bcc-csm1-1 | False | 3.1 | -0.1 | - | 3.7 | 4.9 | 1.8 |
| CCSM4 | False | 2.9 | -0.2 | -4.9 | 4.3 | 4.9 | 1.8 |
| CESM2 | True | 5.2 | - | - | 4.9 | 7.2 | 1.6 |
| CNRM-CM5 | False | 3.3 | 0.1 | -2.7 | 4.0 | 5.1 | 1.7 |
| COSMOS-ASO | False | N/A | - | -5.7 | - | - | 2.2 |
| CSIRO-Mk3-6-0 | False | 4.1 | -0.0 | - | 3.7 | 5.7 | 1.9 |
| CSIRO-Mk3L-1-2 | False | 3.1 | 0.0 | - | 3.2 | - | 1.5 |
| EC-EARTH-2-2 | False | 4.2 | -0.1 | - | - | - | - |
| FGOALS-f3-L | True | 3 | - | - | 5.0 | 5.2 | 1.8 |
| FGOALS-g2 | False | 3.7 | -0.8 | -4.5 | 3.3 | 5.7 | 2.0 |
| FGOALS-s2 | False | 4.5 | -0.1 | - | 4.4 | 6.0 | 2.0 |
| GISS-E2-1-G | True | 2.7 | - | - | 2.0 | 4.2 | 1.2 |
| GISS-E2-R | False | 2.1 | -0.1 | -4.8 | 3.0 | 3.4 | 1.5 |
| HadGEM2-CC | False | 4.5 | 0.2 | - | - | - | - |
| HadGEM2-ES | False | 4.6 | 0.2 | - | 4.6 | 6.2 | 1.1 |
| INM-CM4-8 | True | 2.1 | - | - | 3.0 | 3.6 | 1.7 |
| IPSL-CM5A-LR | False | 4.1 | -0.2 | -4.7 | 4.2 | 6.1 | 2.4 |
| IPSL-CM6A-LR | True | 4.5 | - | - | 5.7 | 7.3 | 1.4 |
| MIROC-ES2L | True | 2.66 | - | - | 3.6 | 4.1 | 1.3 |
| MIROC-ESM | False | 4.7 | -0.6 | -5.3 | 4.1 | 6.0 | 1.9 |
| MPI-ESM-P | False | 3.5 | -0.2 | -4.4 | 4.2 | 5.8 | 1.7 |
| MRI-CGCM3 | False | 2.6 | -0.1 | -4.7 | 3.4 | 4.4 | 2.5 |
| MRI-ESM2-0 | True | 3.1 | - | - | 3.6 | 4.8 | 1.6 |
| NESM3 | True | 3.7 | - | - | 5.9 | 6.9 | 1.7 |

**Table 1.** Details of the models and experiments involved in the analysis. Each experiment provides the global mean change in surface temperature from the preindustrial control simulation ($\Delta$T). The (actual) hydrological sensitivity HS is the global mean percentage change in precipitation divided by the temperature change. It is was calculated via linear regression through all the simulations if available. Where fewer experiments existed it was calculated as the directed average of the values, excluding the midHolocene simulation.



### 2.7.2 The El Niño/Southern Oscillation (ENSO)

The El Niño/Southern Oscillation (Bjerknes, 1966) is an atmosphere-ocean coupled mode of variability with large-scale changes in SST, SLP precipitation and winds as well as the ocean thermocline depth in the equatorial pacific varying semi-periodically with a timescale of 2-10 years (Philander, 1983). ENSO is one of the main drivers of global mean temperature
variability, with global teleconnections (Bjerknes, 1969), and a pronounced impact on the global energy balance and global mean temperature (Trenberth and Fasullo, 2012; Foster and Rahmstorf, 2011). The SLP oscillation in the South Pacific ('Southern Oscillation') was first described by Walker and Bliss E.W. (1932), and the link between atmospheric oscillating patterns and local ocean circulation first described by Bjerknes (1966). Here we use the Niño3.4 and Niñ4 indices, which are the equatorial (5°S–5°N) area-averaged SST anomalies over the regions 170°W–120°W and 160°E–150°W respectively (Trenberth, 1997;
Deser et al., 2010, 2012b). Computations are based on the ncl-script `sst.indices.ncl` (Phillips et al., 2014).

### 2.7.3 The Interdecadal Pacific Oscillation (IPO)

The Interdecadal Pacific Oscillation (IPO) shows a pattern of SST change similar to ENSO (equatorial warming), but with different impacts (Power et al., 1999; Meehl and Hu, 2006). Here we construct a monthly index timeseries based on the first PC of 13-yr low pass filtered Pacific (40°S–60°N, 110°E–70°W) area-weighted SST anomalies, where the global mean SST
anomaly has been removed at each timestep.

### 2.7.4 The Indian Ocean Dipole (IOD)

The Indian Ocean Dipole (IOD) is an irregular pattern of SST variability in the Indian Ocean independent of ENSO in the Pacific (Webster et al., 1999). In an IOD$^-$ event, the western region warms and eastern region cools. The opposing pattern, with a decrease in the zonal temperature gradient is the positive IOD mode (IOD$^+$). The associated changes in surface pressure
and rainfall lead to rainfall modulation and extreme precipitation events at the western/eastern boundaries (Webster et al., 1999). Its subdecadal variability is modulated on decadal to multi-decadal timescales (Ashok et al., 2004). Here, the index time series is calculated using script `sst.indices.ncl` based on the CVDP (Phillips et al., 2014), as the difference of the area-averaged SST anomaly between the regions 50°E–70°E, 10°S – 10°N and 90°E–110°E, 10°S-equator (Saji et al., 1999).

### 2.7.5 The Atlantic Meridional Mode (AMM)

The Atlantic Meridional Mode (AMM), sometimes called the Atlantic dipole mode or gradient mode, is a leading mode of SST variability in the equatorial Atlantic (Servain et al., 1999). The SST pattern with opposing anomalies on either side of the equator modulates the meridional gradient of the sea surface temperature anomaly in the tropical Atlantic, and hence the movement of the Intertropical Convergence Zone (ITCZ) and associated precipitation (Xie and Carton, 2004). The SST gradient is complemented by cross-equatorial atmospheric flow, strengthened by wind-evaporation-surface temperature feedbacks (Xie
and Carton, 2004). The AMM has been linked with hurricane activity in the area (Vimont and Kossin, 2007) and impacts rainfall over tropical Atlantic/NE Brazil/Sahel (Kushnir et al., 2006). Following Doi et al. (2010), the AMM state is defined



here as the basin-wide, area average, detrended SST anomaly difference between the two hemispheres 15–5°N, 50–20°W
minus the average of 15°S–5°S, 20°W–10°E (Phillips et al., 2014).

### 2.7.6    The Atlantic Zonal Mode (ATL3)

Atlantic Zonal Mode (ATL3) is an equatorial coupled mode, similar to ENSO (Zebiak, 1993), therefore sometimes referred to
as 'Atlantic Niño' (Xie and Carton, 2004). Calculation of the mode in the CVDP follows Zebiak (1993), and is based on the
area average of the detrended SST anomaly over the region 3°N - 3°S, 20 - 0°W. The ATL3 displays interannual variations
with roughly a four-year period. Its variations are linked to rainfall variability in the Sahel region (Giannini et al., 2003).

### 2.7.7    The Pacific Decadal Oscillation (PDO)

The Pacific Decadal Oscillation (PDO), also termed Interdecadal Pacific Oscillation (Power et al., 1999), is the leading mode
of variability of monthly SST anomalies over the North Pacific after global mean anomaly is removed. It emerges as a mode
partially driven by ENSO and independent, stochastically emerging variations (Deser et al., 2010; Newman et al., 2003; Mantua
et al., 1997; Schneider and Cornuelle, 2005) with a timescale of decades (Mantua et al., 1997; Deser et al., 2010; Schneider
and Cornuelle, 2005). However, no clear spectral peak has been identified (Deser et al., 2010), as it arises from a superposition
of SST fluctuations with different dynamical origins (Schneider and Cornuelle, 2005; Deser et al., 2010). The PDO was first
described in 1997 as recurring climate pattern of ocean-atmosphere variability over North Pacific and linked to impacts on
Salmon production and coastal surface temperatures on the west coast of the North American continent and the adjacent sea
surface (Mantua et al., 1997). The index is associated with temperature/precipitation changes over western and eastern edges
of North Pacific and displays positive correlation with winter precipitation in California (Mantua et al., 1997). The pattern is
generally similar to ENSO variations but with weaker Southern Pacific imprint (Deser et al., 2010). We calulate a monthly
index time series from the leading principal component of the area-weighted SST anomalies in the box 20–70°N to 110°E–
100°W, where the global mean SST anomaly for each time step has been removed (Deser et al., 2010) based on the script
pdo.ncl from the CVDP (Phillips et al., 2014).

### 2.7.8    North Atlantic Oscillation (NAO) and the Northern Annular Mode (NAM)

The North Atlantic Oscillation (NAO) is a quasi-periodic spatial pattern of sea-level pressure changes between Arctic and
Atlantic (Stephenson et al., 2003; Walker and Bliss E.W., 1932). NAO variations impact the atmospheric circulation over
North Atlantic and the strength of the Westerly inflow into Europe, influencing storm tracks, temperature and precipitation, in
particular in boreal winter (Hurrell, 1995; Hurrell and Deser, 2010). It varies on a seasonal, interannual to decadal timescale
(Hurrell, 1995). In NAO+ phases, a large difference in SLP between the high and mid-latitudes implies a strong SLP gradient
and strong westerly inflow into central Europe. In positive NAO-phases the smaller difference in pressure is associated with a
southerly shift in the North Atlantic storm tracks and enhanced precipitation in the Mediterranean and North Africa. Here we
calculate the NAO index using the script psl.nam_nao.ncl (Phillips et al., 2014), based on the first principal component of





the boreal winter (DJF) area-weighted annual SLP average in the box 20– 80°N, -90 – 40°E (Hurrell and Deser, 2010). Given that this calculation results in a normalised time series, to look at changes in NAO variability we consider the spatial standard deviation of the EOF over the box instead (Power et al., 2013).

The Northern Annular Mode (NAM) describes the zonal SLP gradient between the polar regions and the subtropics. This gradient governs synoptic (5-day-mean) variability of sea-level pressure in the northern hemisphere (Lorenz, 1951). By definition, it is related to the NAO. Here, it is calculated as the leading EOF of the area-weighted monthly-mean SLP anomalies over the latitudes 20–90°N (Hurrell and Deser, 2010), with its variability measured by the spatial standard deviation of this EOF (Power et al., 2013).

**2.7.9   Southern Annular Mode (SAM)**

The Southern Annular Mode (SAM) index gives the strength of the sea-level pressure gradient in the Southern hemisphere mid-latitudes (Karoly, 1990). It is a distinctive pattern of climate variability in the Southern Hemisphere, in particular in winter (Karoly, 1990; Marshall, 2003). The variations in the SLP gradient impact regional temperatures, precipitation (Marshall, 2003; Gillett et al., 2006) as well as the circulation of the Southern Ocean. Negative values of SAM have been linked with weakenings

of the polar vortex, and an icreasing occurrence of hot and dry extremes in Australia (Lim et al., 2019). SAM impacts latitudinal rainfall distribution from the subtropics to Antarctica, with recent trends towards a more positive mode than over the last 1000 years, and links to an Antarctic interior cooling/peninsula warming(Abram et al., 2014). Here, we calculate the PDO variability using the script `psl.sam_psa.ncl` from the CVDP (Phillips et al., 2014). Seasonal/annual PSL averages are formed over the latitudes 20–90°S, and a square root of the cosine of latitude weighting is applied. The leading EOF is considered to give

the pattern for the SAM (Thompson and Wallace, 2000), and spatial standard deviation of this pattern (Power et al., 2013) is used as our measure of its variability.

**2.8   Changes in precipitation extremes**

We investigate the major large-scale patterns of variability associated with precipitation variability across climates. Based on Fig.3 we find that, in many regions, past and future precipitation variability shows opposing signs. We select five regions

with Mediterranean-type climates (Seager et al., 2019): (1) the southwestern tip of South America, (2) southwestern South Africa, (3) southwestern Australia, (4) coastal western North America, and, (5) the western Mediterranean. These regions, in the present, lie between the poleward edge of the winter Hadley cell and equatorward edge of the mid-latitude storm tracks, and have climates characterized by wintertime precipitation and summertime dryness associated with subtropical subsidence, and display substantial interannual variability (Seager et al., 2019).

For each region, model and experiment we (a) calculate the climatological average, annual mean precipitation and as an individual threshold, the interannual standard deviation of local precipitation. We (b) identify where, in the 50-year times-





lice precipitation falls above or below 1 standard deviation and (c) composite sea-level pressure, surface air temperature and precipitation for these extreme precipitation years across all experiments and model simulations.

## 2.9 Timescale-dependence of the variability changes

The power spectrum, $P(\tau)$, of a climate variable describes how its variability is distributed over the timescales $\tau$, with the integral over the entire spectrum yielding the total variance of the signal (Chatfield, 2004). Here we use multitaper power spectrum (Thomson, 1990) with linear detrending, and investigate the area-weighted mean spectra of the local (grid-box) time series. The scaling exponent $\beta$ is used to summarize the scaling relationship of variance with timescale, or equivalently frequency which relates to timescale as $f = 1/\tau$, assuming that the spectrum approximately follows $P(f) \sim \tau^{\beta}$. The scaling exponent $\beta$ is estimated as the linear slope between the logarithm of the power spectral density and the logarithm of timescales; the fit is performed between 4 months to 20 years. Uncorrelated white noise has no autocorrelation, and the scaling exponent is zero ($\beta = 0$). For $\beta > 0$ ($\beta < 0$), the underlying stochastic process displays positive (negative) autocorrelation. Positive autocorrelation for temperature can be expected (Fredriksen and Rypdal, 2016), while precipitation and pressure have lower, or negative values (Fraedrich et al., 2009).

## 3 Results

### 3.1 Hydrological sensitivity across the ensemble

Fig. 1 shows the range of global mean temperature change and precipitation change from the *piControl* simulations. The *lgm* ensemble has a mean temperature anomaly of 4.2 (range of -2.5 to -6) K, and precipitation anomalies range from -6 to -12%/K. The *midHolocene* ensemble shows no large, consistent global mean changes. However, those models that show wetter conditions show positive global mean temperature anomalies. The *1pctCO2* simulations display temperature anomalies from +3 to +7K, and precipitation increases between 3 and 12%. The *abrupt4xCO2* warming simulations are slightly warmer (+4 to +7K) and wetter (+5 to +12%/K). For the entire ensemble we estimate an overall mean HS of 1.73±0.005 (one standard error of the slope) taking into account all models weighted equally. The equilibrium experiments (*lgm* and *midHolocene*) fall consistently on the 2%/K-line (Allen and Ingram, 2002), whereas the transient warming experiments fall below. We find no discernable difference between the precipitation scaling between the CMIP5 and CMIP6 models. We find no systematic relationship between ECS and HS. We note that our findings hold with and without calendar adjustment.

### 3.2 Changes in local interannual variability

Changes in temperature, and temperature variability (Fig. 2). As expected, we find globally cooler conditions for the LGM. These are highly consistent across the ensemble, as the stippling, indicating that at least 2/3rds of the considered models show the same sign as the mean, spreads across the entire field (Fig. 2a). Comparing this to Fig. 2e, which shows the change in simulated temperature variability in the *lgm* experiment vs. the *piControl* as the ratio of standard deviations of the annual





means shows that the interannual temperature variance is high in areas which experienced much colder conditions (at the sea-ice edges), and where the lower sea level led to more exposed shelves (Indonesia) as well as at the edges of the large continental ice sheets (Laurentide, European). The simulated *lgm* temperature variability is higher in the mid-to-high latitudes of both hemispheres, but large areas of the Tropics, in particular the ENSO region, South America, Southern Africa and the

West Pacific Warm Pool show decreases in interannual temperature variance against the *piControl* experiment. Overall, the mean-change pattern of the *lgm* experiment is weakly anticorrelated with the variance-change pattern (r=-0.12, p<0.05 based on area-weighted Pearson correlation and a one-sided t-test conservatively assuming 500 degrees of freedom, accounting for the high degree of spatial autocorrelation in the fields).

The local changes in mean precipitation for the *lgm* simulations (Fig. 3a) are overall negative, consistent with the globally

decreased precipitation (Fig. 1). We find consistent shifts towards higher precipitation in the continental areas of both hemi-spheres affected by subtropical cyclonic precipitation, over northern Africa, southern Africa, across the subtropical southern Atlantic, as well as southwestern North America. Interannual precipitation variance in the *lgm* simulations is lower than in the control simulations with the exception of the areas which have higher mean precipitation, where variability also increases (Fig. 3e). Across the multimodel field, mean and variance change are positively correlated (r=0.63, p<0.01).

The *midHolocene* show weak but consistent (sub)tropical cooling, and moderately warmer conditions in the annual mean temperatures (Fig. 2b), consistent with the positive high-latitude insolation forcing (Sect. 2.3). Overall, the interannual temper-ature variance shows patterns of higher and lower-than-*piControl* variance with modest degrees of inter-model consistency. Similar to the *lgm* variance ratio field, there are reductions in the tropical Atlantic temperature variance, consistent with a local increase in precipitation (Fig. 3b), and precipitation variance (Fig. 3f). Precipitation variance appears lower in the Pacific, and

higher over the Atlantic and Indian Ocean sector, with a strong positive precipitation anomaly over Northern Africa. Mean and variance change are strongly correlated for precipitation (r=0.55, p<0.01), but only weakly correlated for temperature (r=0.09, p<0.05).

Mean temperature change for the *1pctCO2*-scenario is consistently positive with stronger warming over the continents

and amplified warming in the high Northern latitudes (Fig. 2c). Interannual temperature variance (Fig. 3g) shows consistent increases in temperature variability over South-Western North America, South America, Africa, Australia, the Indian Peninsula and China as well as over the North Atlantic, and decreases in temperature variance against *piControl* over Northern North America, Scandinavia, the Tibetan Plateau, Northeast China as well as across the Arctic. Surrounding Antarctica, decreasing temperature variance is observable south of the polar circle, but moderate increases in temperature variance are observable

over East Antarctica. Overall, the mean change and variance change patterns are anticorrelated (r=-0.23, p<0.01), meaning that where we find stronger warming we also observe lower simulated temperature variability.

Mean precipitation change across the *1pctCO2*-ensemble is positive (Fig. 1. Inspecting Fig. 3 indicates, however, that this increase affects primarily the high latitudes and the Equatorial area. In South America no clear change in precipitation is discernible, whereas the Sahel and Arabian Sea are wetter. Mean and variance change fields are positively correlated (r=0.67,

p< 0.01). Patterns of temperature and precipitation changes in the *abrupt4xCO2*-scenario (Fig. 2d and 3d) are highly consistent


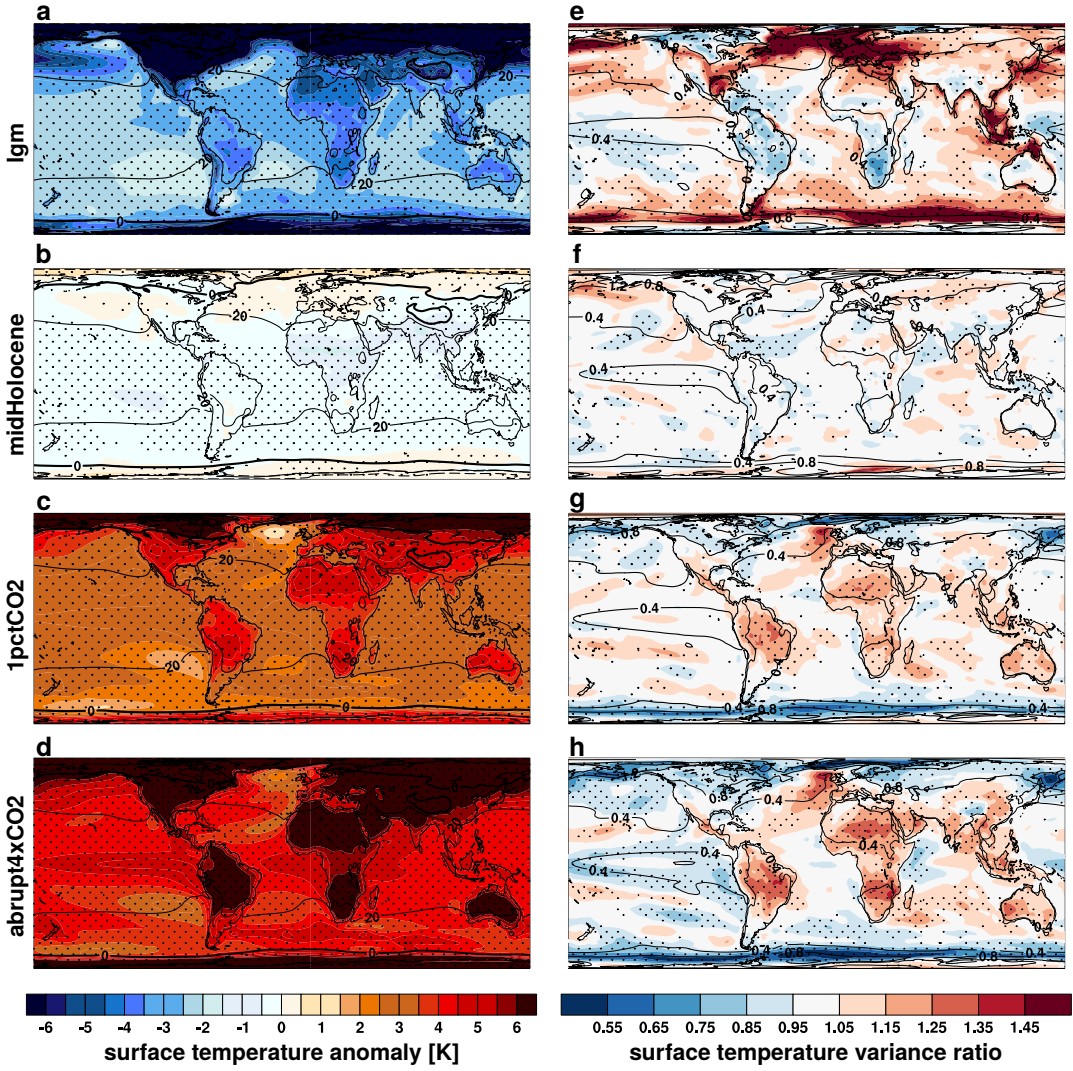

**Figure 2.** The change in mean annual temperature (a-d) and its variability (e-f) across multiple climate experiments. Each panel shows the ensemble average difference. The changes in the mean temperature are calculated as the experiment minus preindustrial control annual means. The changes in variability based on the ratio of the standard deviation of annual mean temperature in the experiment, over that of the *piControl* experiment. Ratios above 1 indicate higher variability in the experiment than in the *piControl*. The contours in each panel show the ensemble-mean pattern in the preindustrial control. Contour variations are due the different number of models available for individual experiments, as the preindustrial ensemble-mean is only computed from models in each experiment. Stippling indicates where the sign of the change agrees for more than 2/3rds of the ensemble.

with those for the *1pctCO2*-scenario (r=0.94,p<0.01 for precipitation, r=0.98, p<0.01 for temperature). In mean and variance, a stronger amplification of the warming patterns (Fig. 2h), over the continents, the North Atlantic, the Indopacific and the areas of the subtropical high are discernible. The polar and continental amplification of the temperature change patterns of the *lgm-*





scenario are mirrored in the areas of warming in the *1pctCO2* and *abrupt4xCO2*-scenarios (r=-0.65 resp. r=-0.64, p<0.01). In particular in the west-coast mid-latitudes where higher precipitation is simulated at the LGM, it appears lower in the warming scenarios of the Northern Hemisphere.

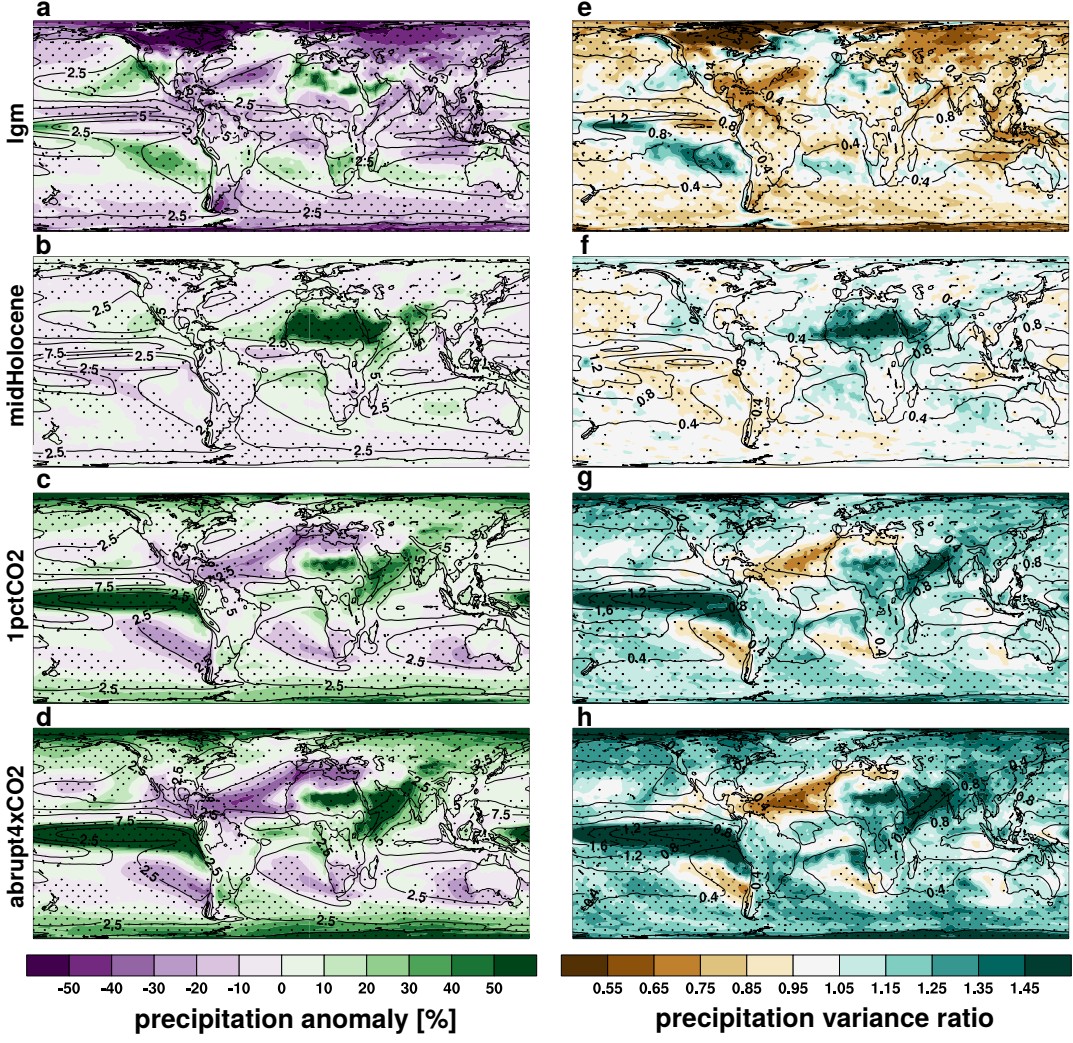

**Figure 3.** The change in mean annual precipitation (a-d) and its variability (e-f) across multiple climate experiments. Each panel shows the ensemble average difference (as percentage changes of the models respective *piControl*). The changes in variability are based on the ratio of the standard deviation of annual mean precipitation in the experiment, over that of the *piControl* experiment. Ratios above 1 indicate higher variability in the experiment than in the *piControl*. The contours in each panel show the ensemble-mean pattern in the preindustrial control (in mm/day). Contour variations are due the different number of models available for individual experiments, as the preindustrial ensemble-mean is only computed from models in each experiment. Stippling indicates where the sign of the change agrees for more than 2/3rds of the ensemble.



### 3.3    Changes in modes of variability

#### 3.3.1    Changes in the global mean

Global mean precipitation increases with global mean temperature across the ensemble (Fig. 1). However, across the multi-model ensemble we find a tendency across the models for the variance of global mean temperature to decrease with the

global mean state, resulting in lower variance than in the *piControl* for the majority of models considered in the idealized warming scenarios and higher-than-preindustrial variance for the *lgm* experiment (Fig. 4a). At the same time, the standard deviation of global mean precipitation increases with approximately 3%/K (Fig. 4b), hence at a higher rate than the global mean precipitation (Fig.1). Comparing these temporal changes against the spatial expression in Figs. 2 and 3 we find that the global reduction of temperature variability with warming is dominated by the ocean and high-latitude signal, whereas the mid-

latitude continental areas show consistent increases in temperature variability with warming. At the same time, the precipitation increase is more inhomogeneous in spatial location and magnitude (Fig. 3d,h).

#### 3.3.2    Changes in SST-based modes

Changes in the SST-based modes of variability across the ensemble are given in Fig. 4c–h. The majority of models (6/9) show a lower-than-preindustrial NINO3.4 and NINO4 standard deviation for the *lgm* and for the *midHolocene* (9/14), and a higher-

than-preindustrial ENSO-index variance for the idealized warming scenarios (Fig. 4c,d). Nonetheless, there is no statistically significant association between global mean temperature and ENSO variability increase (e.g. Christensen et al., 2013). This fits with palaeoENSO restructions of suppressed activity during the mid-Holocene, yet with potential changes in ENSO variability during the LGM (Lu et al., 2018). There are no systematic changes in standard deviation across the ensemble for the PDO (Fig.4e) or the IPO (Fig 4f), although both are not well resolved by the short records analysed here. For the IOD (Fig 4g) there

are no tendencies in the *lgm*-ensemble, with about as many models showing an increased in standard deviation as showing a decrease. However, a majority of models show suppressed IOD activity under the warming scenarios corresponding with the reduced temperature variability over the Arabian Sea upwelling (Fig. 2), which may be a response to the increased ocean stratification seen in the transient simulations (Oyarzún and Brierley, 2019). In the tropical Atlantic, weak but negative trends for the AMM (Fig. 4h) and the ATL3 (Fig. 4i) variability for warmer conditions are found. This fits with the findings of Brierley

and Wainer (2018) and is not inconsistent with the increased future rainfall variability over both the Amazon and West Africa (Fig. 3g,h) - it rather indicates a diminished influence of Atlantic climate variability in the regions.

#### 3.3.3    Changes in atmospheric modes of variability

Let us now consider the atmospheric modes of variability (Fig.4j-l). In the *lgm* experiments, the simulated temperature gradient in the Northern hemisphere is stronger than in the preindustrial - all but one model (Fig. 4) show reduced variability for the

NAM and the NAO. Conversely, in the idealized warming scenarios, with their reduced temperature gradients, more models





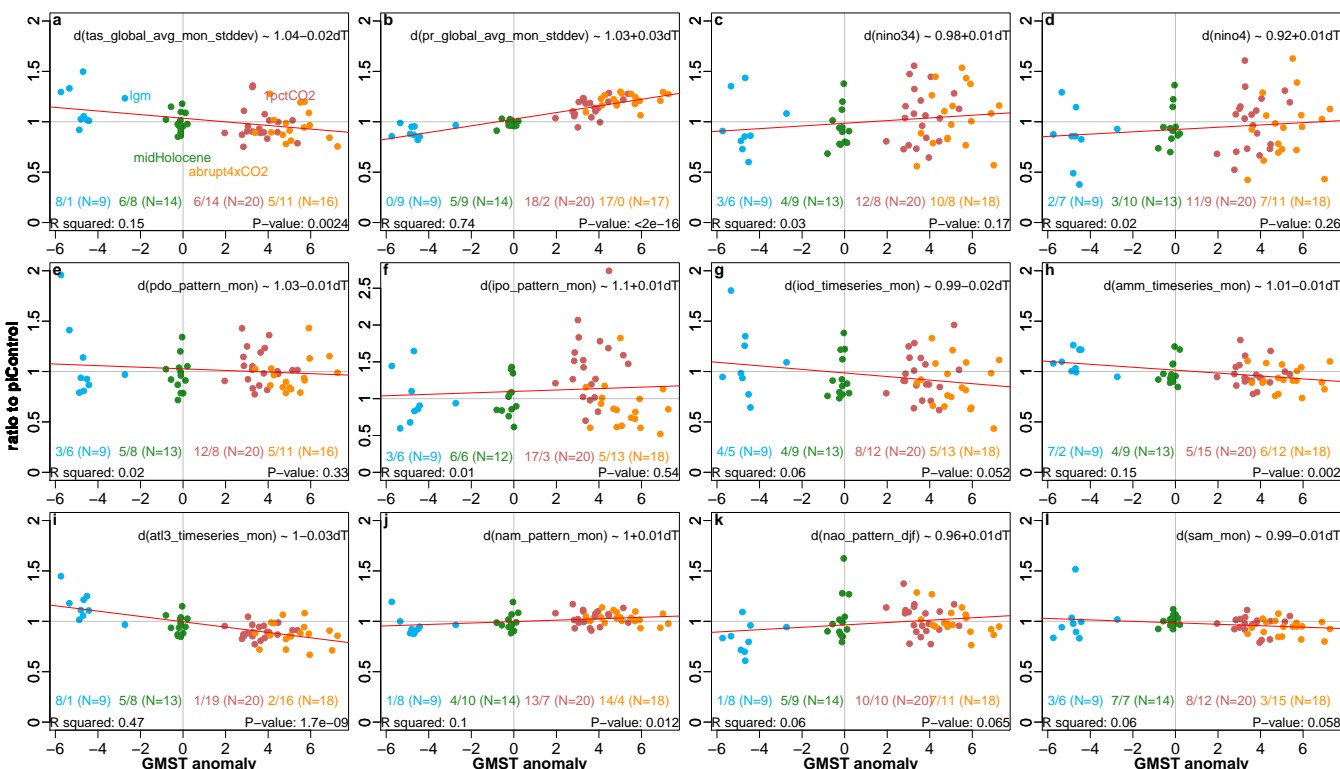

**Figure 4.** Relationship of the standard deviation of climate indices and modes to the change in global mean temperature from preindustrial conditions. Colours indicate the different experiments: CMIP5 and CMIP6 models are not differentiated. (a) Change in the standard deviation of the global, annual mean surface temperature. (b) Change in the standard deviation of the global, annual mean precipitation rate. Changes in the standard deviation (i.e. amplitude of the mode) of (c) ENSO based on the NINO3.4 index and (d) based on the NINO4 region, (e) the PDO, (f) the IPO, (g) the IOD, the meridional (h, AMM) and zonal (i, ATL3) modes of equatorial Atlantic SST variability, and (j) the Northern Annular Mode, (k) the boreal winter NAO and (l) the Southern Annular Mode. All modes are calculated by the Climate Variability Diagnostics Package (Phillips et al., 2014). See Sect. 2.7 for details on the individual modes, and how any changes in mean climate state between the experiments are removed prior to calculation. Linear unweighted fits to the mode changes and the corresponding p-values are given in each panel without censoring for significance.

show increasing standard deviations. Whether a reduced standard deviation indicates a more stable storm track or a more spatially-constrained one requires further investigation and possibly moving away from EOF-based mode definitions.

The Southern Annular Mode shows a tendency towards reduced standard deviations for the idealized warming scenarios (Fig. 4i), but also for the *lgm* experiments. This counter-intuitive response may arise from the competing influences of variabil-

5 ity in the Antarctic sea ice edge (Fig. 2) and the hydrologically-related variabiltiy within the storm tracks (Fig. 3).





### 3.4 Testing the stationarity of modes underlying precipitation extremes

Precipitation changes in Mediterranean-type climates oppose the global mean change across the ensemble. To assess whether the drivers of precipitation in these regions, shown by boxes in the lefthand panels of Fig. 5, are consistent from past to future climates, we investigate sea-level pressure and surface air temperature anomalies associated with high precipitation anomalies
(Fig. 5).

High precipitation years in Patagonia are associated with an increased SLP gradient between the region and the Antarctic continent (Fig. 5a) indicative of positive SAM conditions, a moderate cooling in the South-Eastern Pacific sector and warmer conditions in the South Atlantic and Southern Indian ocean. The reverse situation can be found for years with low precipitation anomalies in Patagonia (SFig 2a,b). The global precipitation composites for regional high and low-precipitation years in SFig. 3
show that years with high-precipitation anomalies in the region are also associated with lower-than-average precipitation in the ENSO regions (SFig 3b). There is no inter-model and inter-experiment consistency in the interannual conditions for high precipitation years in South Africa with regards to SLP and temperature (Fig. 5c,d). The composite plots for Western Australia (Fig. 5e,f) show, similarly to the South American composites, that increased precipitation is found for years with a strong SAM and an increased SLP gradient between Australia and Antarctica. The higher pressure and temperatures in the North Pacific
sector for both Western South American and Western Australian composites could indicate stable teleconnection patterns across the experiments. Cooler conditions prevail throughout the tropics in high-precipitation years, suggesting a decreased southern hemisphere meridional temperature gradient. Precipitation composites (SFig. 3e,f) show a dipole-like structure reminiscent of ENSO, with more precipitation in Western Australia associated with increased precipitation in South-East Asia, and less-than-average precipitation in the Equatorial Pacific.

High precipitation in Western North America is associated with enhanced local low-pressure and higher-than-average SLP over the North Atlantic and Greenland (Fig. 5g) as well as locally warmer conditions (Fig. 5h) and drier conditions to the North and South (Alaska/Mexico, SFig. 3h). These patterns suggest a consistent influence of the PDO and the NAM on interannual precipitation variability in the region.

This is highly similar to the patterns observable for the Western Mediterranean, where high precipitation anomalies are
associated with an increased pressure gradient between the mid- and high latitudes (Fig. 5i), cooler conditions on the Iberian Peninsula and Eurasia and warmer conditions over the Arctic regions of North America and the Labrador Sea (Fig. 5j). For both Western North America and the Western Mediterranean, high annual precipitation years are associated with positive precipitation anomalies in the Equatorial Pacific (SFig. 3h,j).

Therefore, in both South and North America, anomalous precipitation is associated with sea-level pressure variations over
the eastern Pacific in the respective hemisphere (low pressure during wet years, high pressure during dry years) illustrative of circulation patterns that are more or less conducive to water delivery to the continent. In the South, this is also associated with a standing wave structure in surface air temperatures at mid-latitudes, as well as an equatorial Pacific signature reminiscent of ENSO. Precipitation variability over western Australia is also linked to equatorial Pacific temperatures, as well as pressure variations in the Indian and south Pacific oceans, while precipitation variability over the western Mediterranean is clearly linked





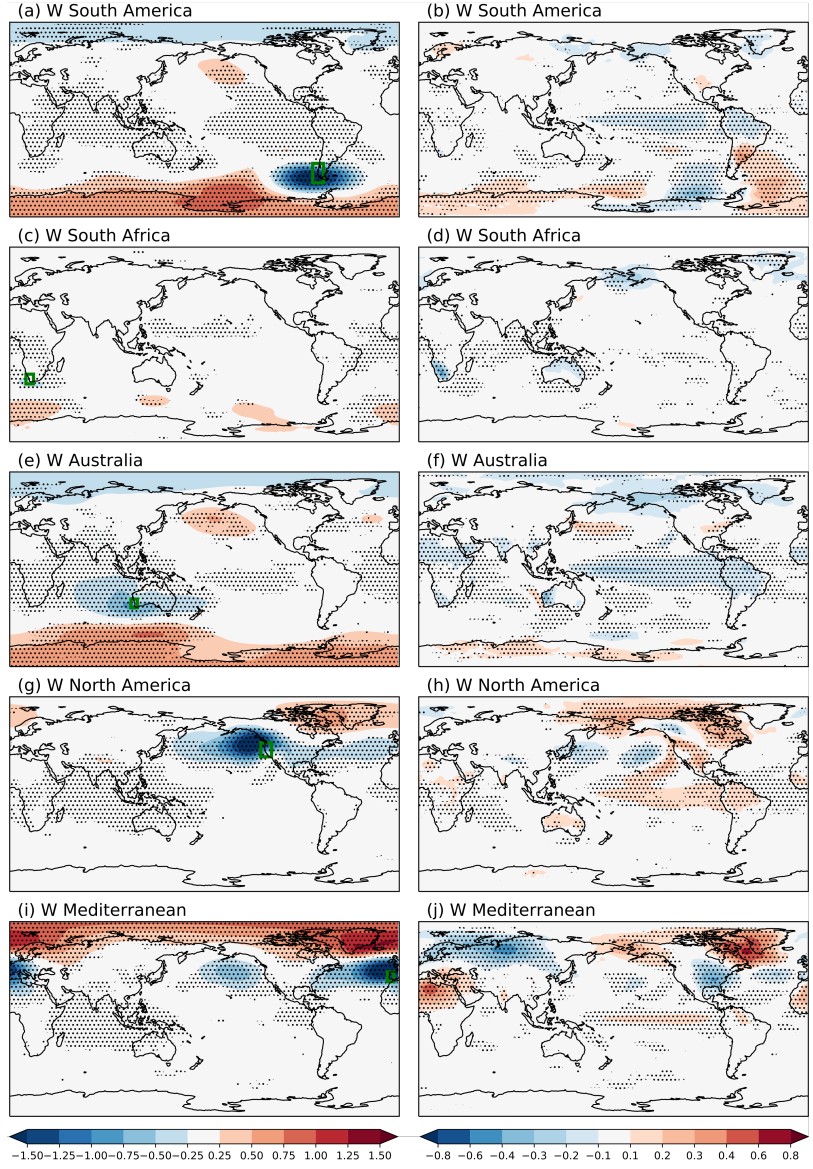

**Figure 5.** Sea-level pressure and surface air temperature anomaly composites for high precipitation years in five regions with Mediterranean climates. Sea-level pressure anomaly composites (panels a,c,e,g,i on the left) and surface air temperature anomaly composites (panels b,d,f,h,j on the right) show the large-scale patterns across models and experiments for years of anomalously high precipitation. In the selected regions, years with one standard deviation above the average were composited for each simulation. Green boxes show the regions of interest on the left-hand panels. Stippling shows regions wherein two-thirds or more of the simulations agree on the sign of the pattern. SFigs. 1 and 2 show the corresponding composites for anomalously low precipitation, and composites for the precipitation change in these years.





to variability over the North Atlantic (likely the NAO), as well as the North Pacific and eastern equatorial Pacific (the latter suggestive of ENSO).

### 3.5   Changes in the spectrum of variability

We investigate the globally averaged, area-weighted power spectra of local monthly temperature (Fig. 6 a,b) and precipitation
(Fig. 6 c,d) anomalies. We find that the spectrum of temperature shows overall higher local temperature variability in the *lgm* experiments, and lower temperature variability for the warm experiments (Fig. 6a), consistent with the findings for total variance (Figs. 2, 3 and 4a,b). Around the ENSO timescale (around 3-5 years), the decrease of variance is less important for the warm experiments, but more important for the *midHolocene* experiment, thus leading to small changes in the scaling before and after that timescale. Overall, the scaling of intraannual to decadal temperature variability is rather consistent for all experiments
(ranging from $\beta$=0.26 to $\beta$=0.35) and changes little with respect to the *piControl* experiment as can also be seen by the rather flat spectral ratio curves (Fig. 6b). The *lgm* curve however shows a small decrease in scaling since the variance increases more on the side of smaller timescales. We also find a remnant annual peak in the idealized warming scenario for the *1pctCO2* temperatures, which could be due to an incomplete detrending of a changing seasonal cycle.

The increase in ENSO-band-variance around the 3-5-year timescale in the warm experiments is more apparent for the
local precipitation anomalies (Fig. 6c) than for temperature, and in addition, it also decreases for the *midHolocene* and *lgm* experiments. Overall, the precipitation variance increases rather consistently over all timescales for the warm experiments with respect to the *piControl* runs, and likewise decreases for the *lgm* and *midHolocene* experiments. The precipitation spectral ratios with respect to the *piControl* simulations (Fig. 6d) outline these patterns clearly. These coherent changes in the global mean spectra are also corroborated by a high degree of consistency in the scaling patterns of surface temperature, precipitation and
surface pressure (SFig. 4), which show 'white', or flat, spectra over the continents and 'red' spectra with variance concentrated at longer timescales over the oceans, particularly along the equator. There is a reddening of the variability over areas where sea-ice is lost in the warm experiments. This could be attributed to the open seas dampening the high-frequency variability more with warming. There is a similar blueing in the *lgm* over the Fram Strait and the Barents Sea where sea-ice cover is extended (SFig. 4). However, there is a reddening over the Arctic for sea-level pressure in the *lgm*.

## 4   Discussion

### 4.1   Changes in climate variability with global mean temperature

Using a wide range of model simulations allowed us to examine the relationship between changes in global mean temperature and climate variability from the perspective of the mean and variance fields, changes in modes of variability, and the timescale-dependency of temperature and precipitation changes. We find that globally averaged temperature variability decreased from
the cold to the warm experiments. This is true for global mean temperature, and the global mean of regional temperatures. We find that changes in temperature variance are more localized than changes in the mean fields. From the cold to the warm(er)

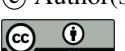



**Figure 6.** Changes of the global mean of the power spectra between the experiments. For temperature (a), variance across all timescales and for most models is highest in the *lgm* experiment, and decreases for the warmer experiments. This is the opposite for precipitation (c), which sees moderate increases in precipitation variability with warming. For each model, we took the ratio of the global mean spectra of each experiment over the *piControl* for both temperature (b) and precipitation (d), thus showing the timescale-dependency of the local variance change. Shaded confidence intervals are based on the entire range of the model ratios.

simulations, temperature variability increases over land, and tends to decrease over the oceans. Temperature variability reduction is particularly strong in the high latitudes, where seasonality and interannual temperature variability are particularly high (Huybers and Curry, 2006). This suggests that changes in temperature variability, in both directions, affect areas which also undergo a large mean-state change.





We find clear indications for shifts in global mean temperature and precipitation variability as well as in tropical Atlantic modes of variability, with the zonal and meridional modes both strongly varying in the *lgm* experiments, and shifting towards weaker variability in the warmer scenarios. This is consistent with the recent findings of Brierley and Wainer (2018), who investigated tropical Atlantic sea surface temperature variability using a similar model ensemble, but also including the histor-

ical era. The zonal gradient mode in the Indian Ocean, IOD, shows a tendency for lower variability in the midHolocene (and thus, for tropical weak cooling) and future warming scenarios, and is therefore not systematically changing with global mean temperature.

The reduced ENSO variability for the *midHolocene* experiments shows that the ENSO mode strength firstly links to the tropical temperature changes, and only secondly to global mean temperature change. This finding is corroborated by the clear

decrease of ENSO-related variance in the global mean spectra for the *midHolocene* experiments. Beyond the ENSO-related timescales, however, changes in temperature and precipitation variability scale across the experiments without strong regard for timescales.

The meridional atmospheric gradient modes of variability in both hemispheres show a weak tendency towards more positive (pole-ward) displacements of the subtropical high with global mean temperature increase in our experiments. This is consistent

with the findings of precipitation reductions in Mediterranean climates at the Western continental edges in both hemispheres. It is, however, unclear to what extent the annular mode (and the westerly jet position) shift due to changes in global mean temperature and the general circulation, or due to ice-sheet height and sea-ice changes that might, to some extent be independent of the change in the mean (Chavaillaz et al., 2013).

## 4.2 Temperature vs. precipitation scaling

We find that, globally averaged, precipitation variability increasing with the global mean temperature of the experiments in our analyses. There is a larger degree of correlation between mean and variability change for precipitation: The drier climates in the *lgm* experiment are spatially extensive, and highly correlated with areas of lower precipitation variability. Conversely, wetter regions in the idealized warming scenarios are also those which show higher precipitation variability. Yet, we find no relationship between the sensitivity of a model to warming under GHG increase, to its propensity for precipitation increase. The

overall scaling of 1.7%/K that we find across the model ensemble is somewhat lower than the 2%/K Li et al. (2013) found for a similar (although somewhat smaller) set of CMIP5 models and experiments and what has been established for earlier models (Allen and Ingram, 2002; Held and Soden, 2006). We note, however, that mean precipitation in the *lgm*- to *midHolocene*-experiments scales with the respective temperature anomalies by 2%/K, and it is the idealized transient warming scenarios that fall below these lines. This could indicate that in these experiments temperature changes are faster than precipitation responses

and, would the experiments be continued, they would get closer to the expected line (Samset et al., 2018; Myhre et al., 2018; Andrews et al., 2010). Indeed, Samset et al. (2018) found that the precipitation increase over the global oceans is markedly slower than that over land, which perhaps explains why we find a scaling that is closer to the terrestrial response in equilibrium experiments (1.8%/K, Li et al. (2013)). Andrews et al. (2010) established that the atmospheric response correlates strongly with the atmospheric component of the radiative forcing, whereas the slow response is, independent of the mechanism leading





to the global temperature change, 2-3%/K. It is unclear, how precipitation variability relates to precipitation extremes, as they operate on much shorter timescales. O'Gorman et al. (2011) found, based on CMIP3 model simulations, that extratropical precipitation extremes increase with 6%/K, and hence at a rate closer to the thermodynamic rate of 7%/K (Allen and Ingram, 2002; Held and Soden, 2006).

## 4.3   Comparison to climate reconstructions and observations

Analysis of instrumental records has shown that the number of record-breaking rainfall events has been increasing over the instrumental era (Lehmann et al., 2015). This is consistent with an ongoing increase in the global mean precipitation rate. Evidence for continental-scale colder/drier conditions at the LGM comes from a variety of terrestrial proxies (Kohfeld and Harrison, 2000; Bartlein et al., 2011), as well as oceanic proxies (MARGO-project-members, 2009). The sampling rate and resolution of proxies for palaeohydrology is, however, often not sufficient to investigate changes in precipitation variability. A high-resolution speleothem record allowed (Luetscher et al., 2015) to relate shifts in the LGM westerly storm tracks in Europe, which are consistent with our finding of enhanced precipitation in the *lgm* experiments. Koutavas and Joanides (2012) suggested that ENSO variability was higher at the LGM than in the Holocene. It is, however, unclear how this relates to our finding of more La-Niña-like conditions in most model simulations, but a reduced ENSO has been corroborated by isotope proxies and isotope-enabled modeling (Zhu et al., 2017). Other studies found ENSO variability to become more persistent with GHG-induced warming (Cai et al., 2014).

## 4.4   Limitations

We have shown that patterns of temperature and precipitation variability in palaeoclimate and future simulations mirror each other, bringing together equilibrium and transient experiments. Nevertheless, there are important limitations that preclude a direct interpretation for future projections (Christensen et al., 2013). Firstly, the snapshots we have been able to analyze are short, and therefore many longer modes of variability are difficult to assess (such as the IOD, or the PDO). Furthermore, we are not able to investigate the variability in the index time series, but only their mean strengtHS. We find that, while temperature variability decreases in the model simulations from the *lgm* to the *1pctCO2* and *abrupt4xCO2* scenarios, the magnitude of change is far lower than that observed in proxy data on longer timescales (Rehfeld et al., 2018). This could be due to models underestimating regional variability beyond the multidecadal timescale (Laepple and Huybers, 2014b; Rehfeld et al., 2018). At the global scale, climate models do, however, capture correct levels of temperature variability (Laepple and Huybers, 2014a; Pages2k-Consortium, 2019).

## 5   Conclusions

We have investigated the simulated changes in surface climate variability across a wide range of climates based on the PMIP3/CMIP5/CMIP6 model ensembles. Across the ensemble, we find global patterns of changes which are roughly opposite between cold (*lgm*) and warm (*1pctCO2/abrupt4xCO2*) experiments. Global mean precipitation increases with temperature



from cold to as-warm-as-preindustrial to warm scenarios. While the simulated temperature variability is higher in the *lgm* scenarios, and decreases globally with temperature, precipitation variability is lower in the cold state, and higher for the warmer scenarios. In both hemispheres, precipitation changes at the mid-latitude western coasts of the continents (California, Patagonia, South Africa, Southern Australia, and the Mediterranean) the inverse of the global mean change in precipitation. They

display more precipitation variability in the *lgm* scenario, and consistently lower precipitation, and precipitation variability, in the *1pctCO2* and *abrupt4xCO2* scenarios. The circulation modes that affect these regions remain consistent across the model ensemble. We investigated, but did not find, an universal relationship between the variability of climate modes and global mean temperature change. No model shows a reduction in temperature variance as large as that for centennial-to-millennial timescales observed in palaeoclimate data for the Last Glacial to Holocene transition, but this could be due to the much shorter

timescales we have investigated here. Yet, on intra-annual to multidecadal timescales, we find evidence of scaling, and that changes in variability appear to occur proportionally across these timescales. Interannual precipitation variability across these simulations appears to robustly, and linearly, relate the relative change in regional variance and the relative change in the mean precipitation. This relationship, and the consistency across timescales, could imply that hydroclimate proxy reconstructions at decadal to centennial timescales provide an additional constraint on simulated past and future precipitation variability changes.

*Code and data availability.* Model data is freely available on the ESGF. The Climate Variability Diagnostic Package is available at http://www.cesm.ucar.edu/working_groups/CVC/cvdp/. Processed data will be made available on the PMIPVarData website at http://www.geog.ucl.ac.uk/ucfaccb/PMIPVarData/. Scripts and code are available on request.

*Competing interests.* The authors declare no competing interests.

*Acknowledgements.* We acknowledge the World Climate Research Programme's Working Group on Coupled Modelling, which is responsi-

ble for CMIP, and we thank the climate modelling groups (listed in Table 1 of this paper) for producing and making available their model output. For CMIP, the US Department of Energy's Program for Climate Model Diagnosis and Intercomparison provides coordinating support and led the development of software infrastructure in partnership with the Global Organization for Earth System Science Portals. K.R. acknowledges funding by the German Research Foundation (DFG, code RE3994-2/1) and the Heidelberg Center for the Environment for providing a venue for discussion. We are thankful for the support of colleagues in two working groups: PAGES Climate Variability Across

Scales (CVAS) and PMIP Past2Future: insights from a constantly varying past. Special thanks are due to Julia Hargreaves, Darrell Kaufman, and Jarmo Kikstra.



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
