# Peer review of "Variability of surface climate in simulations of past and future"

_Earth System Dynamics, 2019_

## Referee Comment (RC1) · Anonymous Referee #1 · 4 Mar 2020

Review of "Variability of surface climate in simulations of past and future" by Rehfeld et al. for consideration in Earth System Dynamics

Overall Comments

The analysis seeks to quantify changes in variability in both temperature and precipitation as a function of the baseline climate state by using simulations from a broad range of climate experiments. The manuscript is well written and logically organized. The core scientific objectives are well articulated. The approach and simulations used are appropriate for the science questions posed. The figures are well-designed generally and nicely illustrate some new and useful results.

My main concerns relate to the summaries provided for the results and the manner

in which the authors gloss over some of what I find to be the the key findings. I also suggest a reconsideration of the figure layouts to more strongly align with the structure of the text (Figs 2,3).

Specific Comments - I recommend splitting the abstract into two paragraphs to make the summary clearer - one for what is done and one for what is found. - p1_10: based on Fig 2, I'm not sure that I find this summary of a decrease in variance for increasing temperature to be true, particularly over land. For example if I am interpreting them correctly, Figs 2g/h show strong positive increases, particularly over land from 40N-S where they are associated with significant impacts. There are various statements in the text that seem at odds with the passage as well (e.g. p12_24-30). Would it make sense to parse this statement a bit more to relate clearly to specific results and distinguish between regions of coherent change that contrast (i.e. land/ocean). - p1_14: is "dominating rainfall variability" appropriate - do they explain the vast majority of variance? across what timescales? - p2_8: the sentence seems to suggest that internal variability is distinct from natural variability? - p2_21: scale linearly? - wording seems to suggest so - p2_30: isn't there also evidence for increases in variability on some timescales? such as ENSO teleconnections? - p3_8: Precipitation changes are also strongly linked inversely to temperature. Wouldn't this therefore be a source of increased temperature variability? There is associated literature on the topic that should be discussed and cited. - p7: For many of these experiments, multiple ensemble members are available. I don't see mention of how many members are used? If only 1, that should be made explicit on page 3. If more than 1, that too should be made explicit and the approach for avoiding overweighting individual models should be described. Despite the additional work, there does seem to be merit in consideration of all available members to address various questions on the role of noise in the results - some listed below. - p8_10: It seems sufficient to merely cite the CVDP rather than each script invoked by it since the CVDP documentation covers this. - p12_14: again referencing the work here that has been done on temperature precipitation relationships seems appropriate. - p12_30: I don't think the global pattern correlation tells the key components of the story. From

40N-40S it seems clear that the PCs are positive. - p13: The caption for Fig 2 should be explicit regarding whether it is the global mean temperature change that is used to compute the ratio or the regional change. - a number of the CVDP variable names are used - which are long and likely not familiar for many readers. I'd recommend creating acronyms for these so that they can be shortened and are more intuitive. - Fig 4: how do you estimate your degrees of freedom in computing P-values? There is mention of 500-DOF in discussion of LGM but that is clearly excessive given the strong mutual dependencies across models, no? Perhaps a more stringent estimate is warranted? - I have a general suggestion regarding the structure of the figures. Since the text is structured to discuss T/P of each experiment why organize the figures to show only T for all experiments and then P for all experiments. Particularly given the mutual relationships that exist, I find merit in having one figure of 4 panels for each experiment - for a total of 4 figures of 4 panels rather than 2 figures of 8. - Fig 4: what is the contribution of internal variability versus model structural contrasts to the scatter in each panel? can multiple ensemble members, where available, be used to estimate a contribution range? I think this would provide key context for interpreting the figure. - Figure 5: I imagine the "W" in the titles corresponds to West? If so I'd spell it out to avoid confusion with "Wet". Also what justifies the selection of the regions? They are much smaller than the climate zones they are intended to represent. Their small size suggests they may be particularly subject to internal variability rather than structural differences across models or experiments. - Figure 6: I suspect that the global mean again masks some important regional effects. How might the results change for land between 40N-S? - p12_13: what is meant by "meridional atmospheric gradient modes of variability"? Is this referring back to results in Section 3.4? Might make this reference more explicit. - p17_11: Is the lack of consistency the result of the choice of such small regions? - p17: There doesn't seem to be any rationale for the organization of paragraphs. Perhaps make one for each region? - p17: After reading Section 3.4 I don't seem to have much of an understanding of the robustness from past to future climate - the stated goal of the section. - p19_5: Why combine land/ocean regions? I think a

distinction should be made, particularly for 40N-S. - p22_14: What is meant by a reduced ENSO? reduced variance in Niño3.4 SSTs? - p22_22: "mean strengtHS"? Note that much of the analysis examined changes in generic variance and then changes in the indices themselves. What is left out is the change in teleconnection strength. Shouldn't this be considered? and isn't it perhaps more important than the changes in the indices themselves? - p22/23: Perhaps cite the figures and panels that support each statement as such references are at times unclear. Some figures seem to clearly contradict the statements made.

---

## Referee Comment (RC2) · Anonymous Referee #2 · 18 Mar 2020

Thank you for asking me to review paper: "Variability of surface climate in simulations of past and future" by Rehfeld et al. Do please accept my apologies for the delay in returning this review.

Any high-quality paper on climate variability is useful, and to say the obvious, it is changes to inter-annual variability that could as much of an effect on society as background climatic changes. This paper provides quite critical information on how the climate system might evolve by a careful scanning across available climate model simulations. The Abstract is clear and captures what the analysis does. The paper builds on what is an under-utilised resource of paleoclimate simulations.

The Reference list is comprehensive, and that in itself makes the paper useful to the climate modelling community.

[Figure]

A few comments:

The research has been undertaken well, and so I can only really offer a few points which the authors might like to consider.

(1) The decrease in local variability as global temperatures increase is always a fascinating feature of the climate system. This reduction also goes against much-perceived wisdom that a warmer world will be a more climatically-volatile world. The authors might like just to note that, possibly in the discussion?

(2) The approach taken is predominantly statistical, which is correct and proper. However, ultimately it would be nice to understand better the background physical processes behind all of the discovered correlations and features. This understanding is not easy when using outputs from climate models developed at research centres across the global, because it can be difficult to "get inside" the models for extra diagnostics. However, a few sentences saying that this analysis could trigger future investigations of the driving processes might help (and possibly with references). For instance, one suggestion is that lower sea-ice coverage in a warmer world will suppress yearly variations in temperature – fitting with the findings of this paper. Other authors have investigated "teleconnections" between the key oceanic forcings and related adjustments to meteorology over land areas. Some of these authors will have offered how atmospheric advection has a role to enforce such connections.

(3) As so much of this paper describes common features between Earth System Models, then maybe at least some sort of mention should be made of the Emergent Constraint (EC) technique? ECs could potentially use the discovered inter-model agreements, in tandem with any additional contemporary measurement, to constrain future projections? Just a sentence or two hinting at this might be useful.

(4) There are substantial sets of paleo measurements that are rarely used by the climate modelling community. Again, maybe for Discussion, but this paper, with its thoughtful aligning of both paleo and future climate simulations, illustrates their huge

potential to constrain climate projection. In other words, if the past can tell us more about the future (e.g. Figure 1, hydrological sensitivity is a valid statistic both for the past and the future), then any past records of simultaneous precipitation and temperature estimates provide valuable extra information.

(5) One thing I especially like about the manuscript is the emphasis on oceanic modes of variability (ENSO, IPO, IOD etc). And this is obviously important given the paper is about variability. The authors will know (i.e. in numerical code) where the boundaries are. Would it be appropriate to give a map somewhere, with each of the oceanic modes of oscillation marked? Most will know where ENSO is, but some of the others are less well known.

(6) Do please work through the paper checking clarity. In general, the manuscript reads well, but in some places, it takes time to fully appreciate the analysis, along with a risk of ambiguity. In addition, the captions should be self-contained. As an example, the Caption for Figure 5, it takes some time to realise that the key point is for each location (as in the subplot headers) corresponds to high rainfall amounts The vague "selected regions" should be expanded more. Or even mark the epic-centre of each region with an annotated arrow for instance.

(7) Some sentences are difficult to read. For instance, in the Conclusions "Global mean precipitation increases with temperature from cold to as-warm-as-preindustrial to warm scenarios.". Maybe better something like: "Modelled global mean precipitation is found to increase as global temperatures also increase. This finding is valid for simulations from pre-industrial periods into a future warmer world, as adjusted by the burning of fossil fuels. However, our paleo-simulations also show this finding to be true, in the transition from colder periods to the warmer period at the beginning of the industrial revolution".

(8) The diagrams are good and informative, but a little attention to formatting and detail could turn them into something exceptional. Just check the basics, such that in each,

all annotation are clear and in sufficiently large font size. Figure 6, make it standard format – so remove the dotted lines maybe?

I am very happy to look at any new version of the manuscript.

---

## Author Comment (AC1) · 21 Apr 2020

**Reply to the reviewers' comments: Variability of Surface Climate in Simulations of Past and Future (esd-2019-92)**

K. Rehfeld, R. Hébert, J. M. Lora, M. Lofverstrom, C. M. Brierley

April 16, 2020

**Summary of changes**

We thank both reviewers for their constructive comments and detailed reading. In response to the suggestions by two reviewers we have

- updated the plots, enhancing Fig. 6 and SFig. 1

- revised text throughout the manuscript to bring out key findings better, differentiate local and global, continental vs. oceanic variability changes and improve clarity

- enhanced the discussion on ENSO changes, temperature-precipitation relationships and potential hydroclimate constraints

- added three new supplementary figures to support the additional discussion

- corrected spelling.

A detailed response to the helpful remarks of the referees is given below.

**1 Reply to the first reviewer**

(Original report cited in italics)
  We thank the reviewer for this positive assessment.

  *My main concerns relate to the summaries provided for the results and the manner in which the authors gloss over some of what I find to be the the key findings. I also suggest a reconsideration of the figure layouts to more strongly align with the structure of the text (Figs 2,3).*
In the revised document we reworked the summary paragraphs and discussion in order to bring out the key findings better. We introduce the comparison across states, that underlies

the layout of our figures earlier (in particular Fig. 2 and 3), which should improve the alignment between visuals and text.

*I recommend splitting the abstract into two paragraphs to make the summary clearer - one for what is done and one for what is found.*
We agree with the reviewer that this improves clarity and have split the abstract accordingly.

*- p1-10: based on Fig 2, I'm not sure that I find this summary of a decrease in variance for increasing temperature to be true, particularly over land. For example if I am interpreting them correctly, Figs 2g/h show strong positive increases, particularly over land from 40N-S where they are associated with significant impacts. There are various statements in the text that seem at odds with the passage as well (e.g. p12-24-30). Would it make sense to parse this statement a bit more to relate clearly to specific results and distinguish between regions of coherent change that contrast (i.e. land/ocean).*

We thank the reviewer for her/his careful inspection of the results. Indeed, temperature variability does not show a uniform decrease globally, and our statements will be made more specific in the revised document. We observe differing trends over continents and oceans at mid-to-low latitudes, with warming associated with increasing temperature variance over the continents of Africa, South America, the maritime continent, Australia and South-East Asia, as well as over South-West Europe and the Southern United States. Decreasing temperature variance is found over the high-latitude continents and the world's oceans except for the Northern Atlantic and over the Indian Ocean. Globally averaged, the local changes point at a decrease in temperature variance. Local changes are generally consistent across timescales, which we demonstrate in the revised document with land/ocean spectra.

*p1-14: is 'dominating rainfall variability' appropriate - do they explain the vast majority of variance? across what timescales?*
Indeed, this is an ambiguous phrase and difficult to pinpoint in models or data. We now clarify that our analysis is at the annual timescale, and to demonstrate that the large-scale modes influencing variability at annual timescales remain stable. We have revised the text in the abstract to better capture the intended meaning: 'By compositing extreme precipitation years across the ensemble, we demonstrate that the same large-scale modes influencing rainfall variability in Mediterranean climates persist throughout palaeoclimate and future simulations.' These modes correspond to extreme precipitation years in the ensemble, identified with composites over the fifty-year time series. Thus, the patterns robustly capture the atmospheric state during annual precipitation extremes (in this case, one standard deviation above/below the mean in annually-averaged precipitation). These are the patterns that emerge, but since this is not an analysis based on Empirical Orthogonal Functions (EOFs) it does not provide the amount of variance it explains. We clarify

this in the revised document.

*- p2-8: the sentence seems to suggest that internal variability is distinct from natural variability?*
That was not our intention and we have revised this sentence. We had meant to use 'internal' when discussing models and 'natural' when discussing the real system.

*- p2-21: scale linearly? - wording seems to suggest so*
We wrote 'In any region, damages do, however, scale with increased variability (Katz and Brown, 1992; Alexander and Perkins, 2013)'. This should imply a direction, but not a qualitative statement on linearity. Depending how damages are estimated (e.g., Katz and Brown (1992) base their statements on a threshold model for crop yields) the increase may follow a different (e.g., exponential) form. Therefore we rephrase this to 'are expected to increase with increasing variability'.

*- p2-30: isn't there also evidence for increases in variability on some timescales? such as ENSO teleconnections?*
We agree with the reviewer that, in the literature, increases in variability have been discussed for specific climate variables, time scales and regions. This includes several studies that suggest ENSO variability may increase (e.g., Cai et al., 2018). We will include a brief discussion of potential ENSO variability changes in the revised document, as the ENSO timescale does show unclear changes (c.f. the power spectra in Fig.6). In particular, it is noticeable that in this power-spectral range some models show a shift in the ENSO frequency, resulting in a peak-and-trough-pattern in the temperature spectral ratio. A change in the ENSO pattern, on the other hand, would not necessarily show up in the spectrum, if the overall variance at the timescale does not change. Amplitude changes are similarly difficult to pinpoint and account for between models. Therefore, for a full discussion of the ENSO changes in CMIP6/PMIP4 we will refer the reader to the ENSO-centered paper currently in open discussion: Brown et al. (2020).

*- p3-8: Precipitation changes are also strongly linked inversely to temperature. Wouldn't this therefore be a source of increased temperature variability? There is associated literature on the topic that should be discussed and cited.*
Indeed, the inverse link between temperature and precipitation has been discussed in the literature (e.g. in the landmark studies of Allen and Ingram, 2002; Adler et al., 2008; Trenberth and Shea, 2005). It is clear that, in particular at daily to interannual timescales soil moisture plays a relevant role in the precipitation feedback on temperature variability (Vidale et al., 2007; Fischer and Knutti, 2013). It is, however, also clear that models have difficulties representing these feedbacks at the land surface, in particular on longer timescales (Rehfeld and Laepple, 2016). The detail of representation of sub-grid-scale

convective processes could also determine whether a local feedback is modeled positively or negatively (Hohenegger et al., 2009). We appreciate the suggestion and add a section on the precipitation-temperature linkage to the discussion.

*- p7: For many of these experiments, multiple ensemble members are available. I don't see mention of how many members are used? If only 1, that should be made explicit on page 3. If more than 1, that too should be made explicit and the approach for avoiding overweighting individual models should be described. Despite the additional work, there does seem to be merit in consideration of all available members to address various questions on the role of noise in the results - some listed below.*
We are sorry if we had failed to specify this. We have only used a single ensemble member for each model (generally r1i1[p1]f1) – this information will be included in the revised document. This approach has been adopted for two reasons. Firstly, it is cleaner, as the reviewer notes it does not overweight individual models in the computation of the ensemble means. Secondly, there is a very low number of the palaeoclimate simulations which have multiple ensemble members.

*- p8-10: It seems sufficient to merely cite the CVDP rather than each script invoked by it since the CVDP documentation covers this.*
OK, the script names were removed from the text.

*- p12-14: again referencing the work here that has been done on temperature precipitation relationships seems appropriate.*
Thank you for the suggestion. We add a reference to the established literature at this point.

*- p12-30: I don't think the global pattern correlation tells the key components of the story. From 40N-40S it seems clear that the PCs are positive.*
Our aim was to provide the global pattern correlations to provide additional support for the ensemble-mean figures that we show. In the revised manuscript we complement the global statements by a refined local view, in particular for the tropical to subtropical land areas. To this effect we have added three new figures to the supplementary material which underline these statements.

*- p13: The caption for Fig 2 should be explicit regarding whether it is the global mean temperature change that is used to compute the ratio or the regional change.*
We rephrased the caption to clarify that we are using the gridbox-scale change for the ratio.

*- a number of the CVDP variable names are used - which are long and likely not fa-*

*miliar for many readers. I'd recommend creating acronyms for these so that they can be shortened and are more intuitive.*

We agree with the reviewer that the CVDP variable names are long. However, in the interest of the reader, we would like to refrain from creating an additional substantial number of acronyms.

*- Fig 4: how do you estimate your degrees of freedom in computing P-values? There is mention of 500-DOF in discussion of LGM but that is clearly excessive given the strong mutual dependencies across models, no? Perhaps a more stringent estimate is warranted?*

Indeed, we assume 500 degrees of freedom for the spatial field pattern correlations (out of a total of $180\times360=64800$ grid boxes) across the fields shown in Fig. 2 and Fig. 3. However, for the regression lines in Fig. 4, which shows the changes in (mode) variability against global mean temperature change in the simulations, we assumed all model simulations and models to be independent. This results in 60 degrees of freedom (with 61 simulations contributing to the regression). We clarify this in the caption in the revision.

*- I have a general suggestion regarding the structure of the figures. Since the text is structured to discuss T/P of each experiment why organize the figures to show only T for all experiments and then P for all experiments. Particularly given the mutual relationships that exist, I find merit in having one figure of 4 panels for each experiment - for a total of 4 figures of 4 panels rather than 2 figures of 8.*

The reviewer is correct, we currently do not follow the subpanel figure order in the discussion. However, our focus is on general relationships across different modeled states. Therefore we add a paragraph in response to this in the beginning of the results section that discusses the idea of comparing relationships across the experiments (from cold to warm, from mean to variance change, from temperature to precipitation). With 4 figures of 4 panels it is more difficult to make out the similarities/differences across the experiments and variables.

*- Fig 4: what is the contribution of internal variability versus model structural contrasts to the scatter in each panel? can multiple ensemble members, where available, be used to estimate a contribution range? I think this would provide key context for interpreting the figure.*

We appreciate the suggestion. Multiple ensemble members are, unfortunately, not available for the palaeoclimate simulations. Nevertheless, we agree that the contribution of internal variability is an important factor to consider. Therefore we utilize the long preindustrial control experiments to estimate the contribution of internal variability. This is then added to Fig. 4 as confidence intervals around the unity line.

*- Figure 5: I imagine the "W" in the titles corresponds to West? If so I'd spell it out to avoid confusion with "Wet". Also what justifies the selection of the regions? They are much smaller than the climate zones they are intended to represent. Their small size suggests they may be particularly subject to internal variability rather than structural differences across models or experiments.*

We have revised the figure titles as suggested. The regions are based on the Köppen climate classification of Mediterranean climates, and in particular the western boundaries of continents wherein the extratropical climate appears to cause precipitation anomalies of different signs than the global mean change from the pre-industrial (see Fig. 3). The selected boxes are actually larger than these regions strictly defined (and in all cases encompass multiple grid boxes), and our analysis in fact shows that the same modes of variability are important across different climate states. In the revised document we add supplementary figures that show that the circulation patterns are robust across climates (except for Western South Africa, where there is no signal).

*- Figure 6: I suspect that the global mean again masks some important regional effects. How might the results change for land between 40N-S?* Indeed, as suspected by the reviewer and as the total variance changes in Fig. 2 in the manuscript make clear, the global mean variance change differs from that over the low-to-mid-latitude land areas. Averaging the spectra over land areas between 40S and 40N we have less clear changes, and most importantly find indications for higher temperature variance in the warm experiments than in the preindustrial control. There is, however, also slightly more temperature variance over these areas in the LGM (cold) experiment than in the preindustrial control across the spectrum. Definitive statements are complicated by the fact that there is less intermodel agreement. We expanded the results section, and the discussion to take this into account. To support this discussion we added three figures to the supplementary material which show the power spectra over land areas, globally and from 40S to 40N, as well as the ocean average to support the discussion of Fig. 6.

*- p12-13: what is meant by "meridional atmospheric gradient modes of variability"? Is this referring back to results in Section 3.4? Might make this reference more explicit.*
Indeed, this is a reference to SAM/NAM and NAO results in Sect. 3.4. We make this more clear in the revised document.

*- p17-11: Is the lack of consistency the result of the choice of such small regions?*
The lack of consistency only occurs for the South African case. This suggests that the Mediterranean regions are generally appropriately sized. Please refer to the response above regarding Fig. 5.

*- p17: There doesn't seem to be any rationale for the organization of paragraphs. Per-*

*haps make one for each region? - p17: After reading Section 3.4 I don't seem to have much of an understanding of the robustness from past to future climate - the stated goal of the section.*

We agree, the previous section title was misleading. We have changed this and revised the text to better explain the results. In addition, we have included additional supplementary figures that better illustrate the robustness of the relevant patterns from past to future climates (with the exception of South Africa). The organization of this section is based on that of the relevant figure, and proceeds between the regions from southwestern South America to the western Mediterranean.

*- p19-5: Why combine land/ocean regions? I think a distinction should be made, particularly for 40N-S.*

We agree with the reviewer that a distinction enhances the discussion and therefore added three new supplementary figures as indicated in the response above.

*- p22-14: What is meant by a reduced ENSO? reduced variance in Niño3.4 SSTs?*

Yes. We shall specify that in the revised manuscript.

*- p22-22: "mean strengtHS"? Note that much of the analysis examined changes in generic variance and then changes in the indices themselves. What is left out is the change in teleconnection strength. Shouldn't this be considered? and isn't it perhaps more important than the changes in the indices themselves?*

We understand the reviewer's comment and sympathise with its sentiment. However, analysing the changes in the teleconnection patterns is not trivial and requires specifying certain decisions that may not be appropriate for all modes. We feel that to do so rigorously could require an individual manuscript for each mode, and therefore is not appropriate for this paper. We further explain this issue using ENSO. The teleconnection changes in these simulations have been explored in Brown et al. (2020) ENSO teleconnections are often computed using composites - there were serious issues dealing with the changes in the mean state conflating with the teleconnections changes (after Cai et al. (2014)). Additionally ENSO teleconnections may not be reciprocal for the different phases.

*- p22/23: Perhaps cite the figures and panels that support each statement as such references are at times unclear. Some figures seem to clearly contradict the statements made.*

We worked through the manuscript again to ensure that the statements cover both the global and the regional scale to avoid misunderstandings

**References**

Adler, R. F., Gu, G., Wang, J.-J., Huffman, G. J., Curtis, S., and Bolvin, D.: Relationships between global precipitation and surface temperature on interannual and longer timescales (1979–2006), Journal of Geophysical Research, 113, D22 104, https://doi.org/10.1029/2008JD010536, URL `http://doi.wiley.com/10.1029/2008JD010536`, 2008.

Allen, M. R. and Ingram, W. J.: Constraints on future changes in climate and the hydrologic cycle, Nature, 419, 228–232, https://doi.org/10.1038/nature01092, 2002.

Brown, J., Brierley, C., An, S.-I., Guarino, M.-V., Stevenson, S., Williams, C., Zhang, Q., Zhao, A., Braconnot, P., Brady, E., Chandan, D., D'Agostino, R., Guo, C., LeGrande, A., Lohmann, G., Morozova, P., Ohgaito, R., O'ishi, R., Otto-Bliesner, B., Peltier, R., Shi, X., Sime, L., Volodin, E., Zhang, Z., and Weipeng, Z.: Comparison of past and future simulations of ENSO in CMIP5/PMIP3 and CMIP6/PMIP4 models, Clim. Past Discuss., https://doi.org/10.5194/cp-2019-155, 2020.

Cai, W., Borlace, S., Lengaigne, M., van Rensch, P., Collins, M., Vecchi, G., Timmermann, A., Santoso, A., McPhaden, M. J., Wu, L., England, M. H., Wang, G., Guilyardi, E., and Jin, F.-F.: Increasing frequency of extreme El Nino events due to greenhouse warming, Nature Climate Change, 4, 111–116, https://doi.org/10.1038/nclimate2100, 2014.

Cai, W., Wang, G., Dewitte, B., Wu, L., Santoso, A., Takahashi, K., Yang, Y., Carréric, A., and McPhaden, M. J.: Increased variability of eastern Pacific El Nino under greenhouse warming, Nature, 564, 201–206, https://doi.org/10.1038/s41586-018-0776-9, 2018.

Fischer, E. M. and Knutti, R.: Robust projections of combined humidity and temperature extremes, Nature Climate Change, 3, 126–130, https://doi.org/10.1038/nclimate1682, 2013.

Hohenegger, C., Brockhaus, P., Bretherton, C. S., and Schär, C.: The soil moisture-precipitation feedback in simulations with explicit and parameterized convection, Journal of Climate, 22, 5003–5020, https://doi.org/10.1175/2009JCLI2604.1, 2009.

Katz, R. W. and Brown, B. G.: Extreme events in a changing climate: Variability is more important than averages, Climatic Change, 21, 289–302, https://doi.org/10.1007/BF00139728, 1992.

Rehfeld, K. and Laepple, T.: Warmer and wetter or warmer and dryer? Observed versus simulated covariability of Holocene temperature and rainfall in Asia, Earth and Planetary Science Letters, 436, 1–9, https://doi.org/10.1016/j.epsl.2015.12.020, 2016.

Trenberth, K. E. and Shea, D. J.: Relationships between precipitation and surface temperature, Geophysical Research Letters, 32, 2–5, https://doi.org/10.1029/2005GL022760, 2005.

Vidale, P. L., Lüthi, D., Wegmann, R., and Schär, C.: European summer climate variability in a heterogeneous multi-model ensemble, Climatic Change, 81, 209–232, https://doi.org/10.1007/s10584-006-9218-z, 2007.

---

## Author Comment (AC2) · 21 Apr 2020

**Reply to the reviewers' comments: Variability of Surface Climate in Simulations of Past and Future (esd-2019-92)**

K. Rehfeld, R. Hébert, J. M. Lora, M. Lofverstrom, C. M. Brierley

April 16, 2020

**Summary of changes**

We thank both reviewers for their constructive comments and detailed reading. In response to the suggestions by two reviewers we have

- updated the plots, enhancing Fig. 6 and SFig. 1

- revised text throughout the manuscript to bring out key findings better, differentiate local and global, continental vs. oceanic variability changes and improve clarity

- enhanced the discussion on ENSO changes, temperature-precipitation relationships and potential hydroclimate constraints

- added three new supplementary figures to support the additional discussion

- corrected spelling.

A detailed response to the helpful remarks of the referees is given below.

**1    Reply to the second reviewer**

(Original report cited in italics)

*Thank you for asking me to review paper: "Variability of surface climate in simulations of past and future" by Rehfeld et al. Do please accept my apologies for the delay in returning this review. Any high-quality paper on climate variability is useful, and to say the obvious, it is changes to inter-annual variability that could as much of an effect on society as background climatic changes. This paper provides quite critical information on how the climate system might evolve by a careful scanning across available climate model simulations. The Abstract is clear and captures what the analysis does. The paper builds on what is an under-utilised resource of paleoclimate simulations. The Reference list is*

*comprehensive, and that in itself makes the paper useful to the climate modelling community.*

We thank the reviewer for this positive assessment.

*A few comments: The research has been undertaken well, and so I can only really offer a few points which the authors might like to consider. (1) The decrease in local variability as global temperatures increase is always a fascinating feature of the climate system. This reduction also goes against much-perceived wisdom that a warmer world will be a more climatically-volatile world. The authors might like just to note that, possibly in the discussion?*

Indeed, the findings at the global scale do contradict the intuitive expectation (rooted in the molecular physics of gases, perhaps, in the Maxwell-Boltzmann-distribution?). In the revised manuscript we will differentiate more strongly between the global and regional scales in the discussion, and will take up this suggestion.

*(2) The approach taken is predominantly statistical, which is correct and proper. However, ultimately it would be nice to understand better the background physical processes behind all of the discovered correlations and features. This understanding is not easy when using outputs from climate models developed at research centres across the global, because it can be difficult to "get inside" the models for extra diagnostics. However, a few sentences saying that this analysis could trigger future investigations of the driving processes might help (and possibly with references). For instance, one suggestion is that lower sea-ice coverage in a warmer world will suppress yearly variations in temperature – fitting with the findings of this paper. Other authors have investigated "teleconnections" between the key oceanic forcings and related adjustments to meteorology over land areas. Some of these authors will have offered how atmospheric advection has a role to enforce such connections.*

We absolutely agree with the reviewer in that a better understanding of the physical mechanisms of changing climate variability is crucial to understand our results. Some research on this exists, but a conclusive view across regions, seasons and timescales is difficult. On interannual timescales, sea-ice extent has been shown to correlate with global temperature variability (Huntingford et al., 2013). However, it remains unclear whether this would remain to be the case if a summer-ice-free Arctic has been reached, and how it influences low-latitude climate variability. A key role from the seasonal (Holmes et al., 2016) to the millennial (Rehfeld et al., 2018) timescale is certainly played by the meridional temperature gradients that modulate atmospheric flows. However, due to the turbulent nature of the atmosphere changes to the contributions of latent and sensible heat transport to mid-to-high latitude temperature variability are difficult to assess (Schneider et al., 2015). Therefore, as the reviewer notes, better understanding of the background physical processes behind the correlations and features is required. Our analysis and results therefore clearly calls for extending future research on the driving processes of variability changes. We add this

to the discussion and conclusion of the manuscript.

*3) As so much of this paper describes common features between Earth System Models, then maybe at least some sort of mention should be made of the Emergent Constraint (EC) technique? ECs could potentially use the discovered inter-model agreements, in tandem with any additional contemporary measurement, to constrain future projections? Just a sentence or two hinting at this might be useful.*

This a good suggestion that we will adopt in the revised manuscript. There have so far been few examples of variability-based observational constraints (e.g., Cox et al., 2018). We will add this idea as motivation in the introduction, and then return to the theme in the discussion.

*(4) There are substantial sets of paleo measurements that are rarely used by the climate modelling community. Again, maybe for Discussion, but this paper, with its thoughtful aligning of both paleo and future climate simulations, illustrates their huge potential to constrain climate projection. In other words, if the past can tell us more about the future (e.g. Figure 1, hydrological sensitivity is a valid statistic both for the past and the future), then any past records of simultaneous precipitation and temperature estimates provide valuable extra information.*

Again this is very useful comment. It is something that we have started thinking seriously about. Highlighting the potential in our revised discussion will not only make the manuscript stronger, but help motivate our own future research. There are some methodological issues that need to be resolved before it can be deployed in earnest though. Crucially, obtaining joint (or closeby) and robust estimates of temperature and precipitation from proxy data is a fundamental challenge (Rehfeld and Laepple, 2016; Rehfeld et al., 2016).

*(5) One thing I especially like about the manuscript is the emphasis on oceanic modes of variability (ENSO, IPO, IOD etc). And this is obviously important given the paper is about variability. The authors will know (i.e. in numerical code) where the boundaries are. Would it be appropriate to give a map somewhere, with each of the oceanic modes of oscillation marked? Most will know where ENSO is, but some of the others are less well known.*

We agree with the reviewer that both atmospheric and oceanic modes of variability are important to consider. We have provided Supplementary Fig. 1 with the boundaries of the modes, and will highlight it in the manuscript revision.

*(6) Do please work through the paper checking clarity. In general, the manuscript reads well, but in some places, it takes time to fully appreciate the analysis, along with a risk of ambiguity. In addition, the captions should be self-contained. As an example, the Caption*

*for Figure 5, it takes some time to realise that the key point is for each location (as in the subplot headers) corresponds to high rainfall amounts The vague "selected regions" should be expanded more. Or even mark the epic-centre of each region with an annotated arrow for instance.*

We have modified the caption. We also note that here, as in the original submission, the regions are marked by green boxes. Furthermore, we worked through the manuscript again to ensure each figure/caption is more self-explanatory.

*(7) Some sentences are difficult to read. For instance, in the Conclusions "Global mean precipitation increases with temperature from cold to as-warm-as-preindustrial to warm scenarios.". Maybe better something like: "Modelled global mean precipitation is found to increase as global temperatures also increase. This finding is valid for simulations from pre-industrial periods into a future warmer world, as adjusted by the burning of fossil fuels. However, our paleo-simulations also show this finding to be true, in the transition from colder periods to the warmer period at the beginning of the industrial revolution".*

We thank the reviewer for the detailed reading and this suggestion. We will re-phrase this sentence in the revision and are checking through the entire manuscript again to ensure more clarity.

*(8) The diagrams are good and informative, but a little attention to formatting and detail could turn them into something exceptional. Just check the basics, such that in each, all annotation are clear and in sufficiently large font size. Figure 6, make it standard format - so remove the dotted lines maybe?*

Thank you for this suggestion. The revised version of Fig. 6 follows a more standard aspect ratio, includes boxes around the panels and consistent label sizes.

**References**

Cox, P. M., Huntingford, C., and Williamson, M. S.: Emergent constraint on equilibrium climate sensitivity from global temperature variability, Nature, 553, 319–322, https://doi.org/10.1038/nature25450, 2018.

Holmes, C. R., Woollings, T., Hawkins, E., and de Vries, H.: Robust Future Changes in Temperature Variability under Greenhouse Gas Forcing and the Relationship with Thermal Advection, Journal of Climate, 29, 2221–2236, https://doi.org/10.1175/JCLI-D-14-00735.1, 2016.

Huntingford, C., Jones, P. D., Livina, V. N., Lenton, T. M., and Cox, P. M.: No increase in global temperature variability despite changing regional patterns., Nature, 500, 327–30, https://doi.org/10.1038/nature12310, 2013.

Rehfeld, K. and Laepple, T.: Warmer and wetter or warmer and dryer? Observed versus simulated covariability of Holocene temperature and rainfall in Asia, Earth and Planetary Science Letters, 436, 1–9, https://doi.org/10.1016/j.epsl.2015.12.020, 2016.

Rehfeld, K., Trachsel, M., Telford, R. R. J., and Laepple, T.: Assessing performance and seasonal bias of pollen-based climate reconstructions in a perfect model world, Climate of the Past, 12, 2255–2270, https://doi.org/10.5194/cp-12-2255-2016, 2016.

Rehfeld, K., Münch, T., Ho, S. L., and Laepple, T.: Global patterns of declining temperature variability from the Last Glacial Maximum to the Holocene, Nature, 554, 356–359, https://doi.org/10.1038/nature25454, 2018.

Schneider, T., Bischoff, T., and Płotka, H.: Physics of Changes in Synoptic Midlatitude Temperature Variability, Journal of Climate, 28, 2312–2331, https://doi.org/10.1175/JCLI-D-14-00632.1, 2015.